# Retinoic acid-induced protein 14 links mechanical forces to Hippo signaling

Wonyoung Jeong [1], Hyeryun Kwon[1], Sang Ki Park [2], In-Seob Lee [3]✉ & Eek-hoon Jho [1]✉

## Abstract

**Cells sense and respond to various mechanical forces from the extracellular matrix primarily by modulating the actin cytoskeleton. Mechanical forces can be translated into biochemical signals in a process called mechanotransduction. Yes-associated protein (YAP) is an effector of Hippo signaling and a mediator of mechanotransduction, but how mechanical forces regulate Hippo signaling is still an open question. We propose that retinoic acid-induced protein 14 (RAI14) responds to mechanical forces and regulates Hippo signaling. RAI14 positively regulates the activity of YAP. RAI14 interacts with NF2, a key component of the Hippo pathway, and the interaction occurs on filamentous actin. When mechanical forces are kept low in cells, NF2 dissociates from RAI14 and filamentous actin, resulting in increased interactions with LATS1 and activation of the Hippo pathway. Clinical data show that tissue stiffness and expression of RAI14 and YAP are upregulated in tumor tissues and that RAI14 is strongly associated with adverse outcome in patients with gastric cancer. Our data suggest that RAI14 links mechanotransduction with Hippo signaling and mediates Hippo-related biological functions such as cancer progression.**

**Keywords** Hippo Signaling; RAI14; Mechanotransduction; Gastric Cancer
**Subject Categories** Cancer; Cell Adhesion, Polarity & Cytoskeleton; Signal Transduction

## Introduction

Cells are constitutively exposed to and respond to various external stimuli to maintain cellular homeostasis. For example, soluble factors such as hormones and growth factors influence cells to promote growth, proliferation and other cellular functions. In addition to soluble factors, cells are also exposed to mechanical forces. Shear stress induced by a fluid or matrix stiffness induced by the extracellular matrix (ECM) generate mechanical forces in cells (Humphrey et al, 2014; Vining and Mooney, 2017). Cells respond to these mechanical forces via plasma membrane proteins, known as mechanosensors, to generate intracellular tension primarily by modulating the actin cytoskeleton, thereby adjusting cellular structure and function (Humphrey et al, 2014). Specifically, when cells are placed in a stiff matrix, seeded under sparse conditions, or attached to a large adhesive region, the mechanical forces applied to the cells remain high. Under these conditions, the interaction with integrin and ECM is promoted, and the formation of focal adhesion complexes (FACs) by the cytoplasmic domain of integrin is facilitated. FACs are a family of protein complexes composed of vinculin, talin, paxillin, focal adhesion kinases (FAKs) and other proteins. The formation of FACs causes the serial activation of FAKs and Rho-associated kinase (ROCK), leading to the assembly of filamentous actin (F-actin). This promotes cellular functions such as proliferation, osteoblast differentiation and fibrosis. In contrast, when cells are placed in a soft matrix, seeded under dense conditions or attached to a small adhesive area, mechanical forces remain low. It inhibits the formation of FACs, the activity of FAKs and the assembly of F-actin. As a result, cellular functions such as apoptosis and adipocyte differentiation are promoted (Humphrey et al, 2014; Panciera et al, 2017).

The mechanical forces induced by F-actin assembly can be translated into biochemical signals that regulate the expression of genes to modulate cellular functions. This series of processes is known as "mechanotransduction" (Humphrey et al, 2014). Among several proteins, Yes-associated protein (YAP) and its paralogue transcriptional co-activator with PDZ-binding motif (TAZ) have been identified as core transcriptional co-activators for mechanotransduction (Dupont et al, 2011). When cells are exposed to high mechanical forces, such as stiff ECM, YAP/TAZ is translocated to the nucleus and promotes the expression of target genes. In contrast, under low mechanical forces, YAP/TAZ is sequestered in the cytoplasm and the expression of its target genes is inhibited.

The Hippo signaling pathway, which is conserved from *Drosophila* to mammals and is a known regulator of YAP/TAZ, plays an important role in several biological functions, including organ size control, development, tissue regeneration, stem cell pluripotency and immunity (Driskill and Pan, 2023; Zheng and Pan, 2019). Dysregulation of Hippo signaling leads to YAP/TAZ dysfunction, which is implicated in several diseases, including cancer and degenerative diseases (Franklin et al, 2023; Piccolo et al, 2023). Mechanistically, activation of the Hippo pathway leads to the sequential phosphorylation and activation of mammalian STE20-like protein kinase 1/2 (MST1/2) and large tumor suppressor kinases 1/2 (LATS1/2). Activated LATS1/2 phosphorylates YAP/TAZ primarily at serine

[1]Department of Life Science, University of Seoul, Seoul 02504, Republic of Korea. [2]Department of Life Sciences, Pohang University of Science and Technology, Pohang 37673, Republic of Korea. [3]Department of Surgery, Asan Medical Center, University of Ulsan College of Medicine, Seoul 05505, Republic of Korea. ✉E-mail: inseoblee77@amc.seoul.kr; ej70@uos.ac.kr

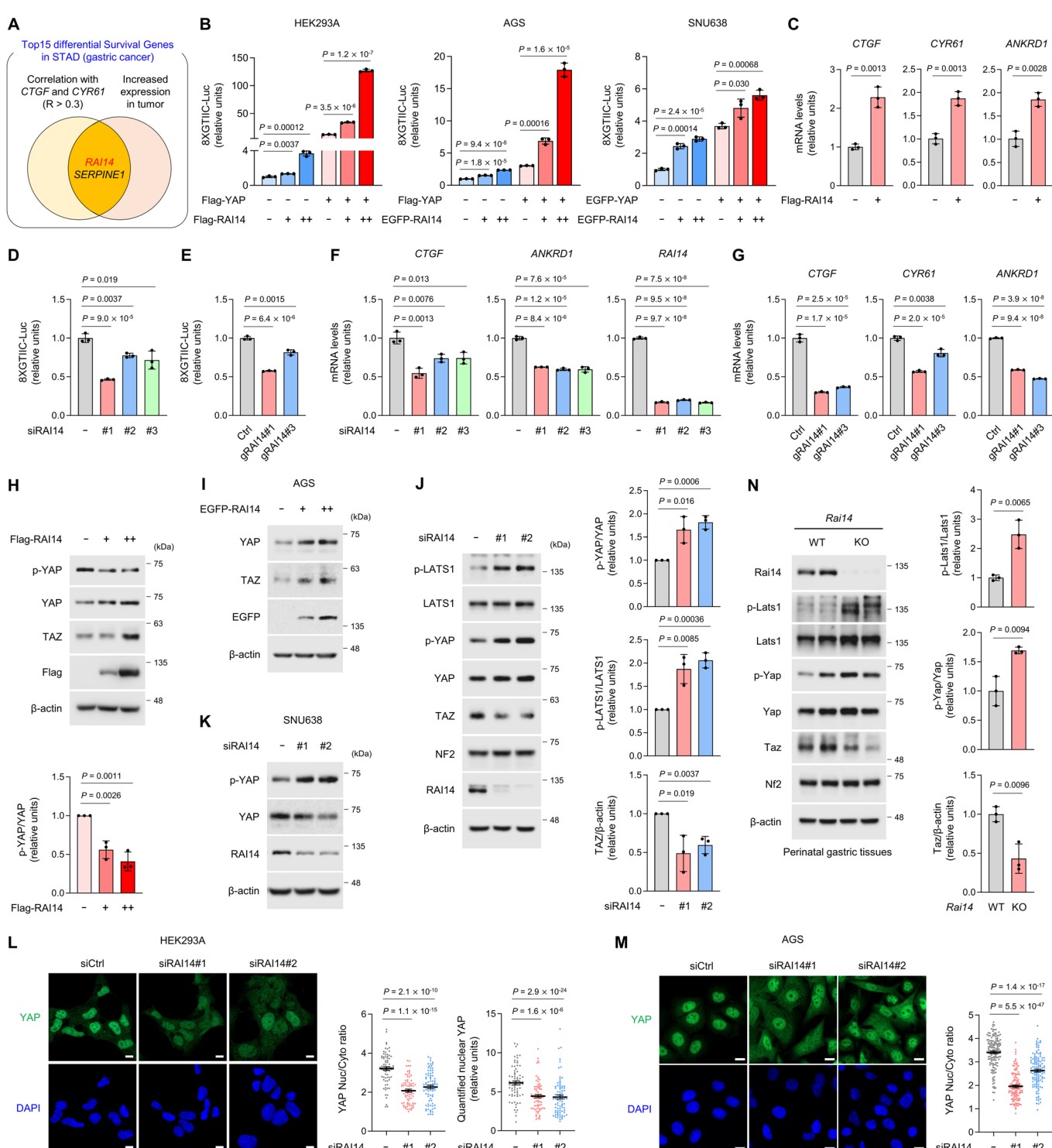

127 (serine 89 in TAZ), leading to cytoplasmic retention or proteasomal degradation of YAP/TAZ. As a result, the transcriptional activity of YAP/TAZ is kept low. In contrast, MST1/2 and LATS1/2 are inactivated upon inhibition of the Hippo pathway, resulting in the YAP/TAZ unphosphorylated. As a result, YAP/TAZ can translocate to the nucleus and interact with TEA domain transcription factors (TEADs) to induce transcription of target genes (Kwon et al, 2022).

There has been progress in elucidating the relationship between mechanical forces, Hippo signaling and YAP/TAZ (Dasgupta and McCollum, 2019; Panciera et al, 2017). However, the detailed mechanisms of how mechanical forces induced by F-actin assembly regulate YAP/TAZ activity remain unclear.

Here, we report that retinoic acid-induced protein 14 (RAI14) responds to mechanical forces and regulates Hippo signaling.

**Figure 1.   RAI14 activates YAP/TAZ.**

(A) Schematic representation of Fig. EV1A as Venn diagram. *RAI14* and *SERPINE1* fulfill all criteria. (B) Overexpression of RAI14 activates YAP-reporter activity. (C) Overexpression of RAI14 in HEK293A cells upregulates the expression of YAP-target genes. (D, E) Downregulation of RAI14 by siRNA (D) or gRNA (E) in HEK293A cells reduces YAP-reporter activity. Lentivirus-transduced HEK293A cells were used as control. (F, G) Downregulation of RAI14 by siRNAs (F) or gRNAs (G) in HEK293A cells reduces the expression of YAP-target genes. (H, I) Overexpression of RAI14 in HEK293A (H) and AGS (I) cells decreases the phosphorylation levels of YAP and increase the protein levels of YAP/TAZ. The p-YAP/total-YAP ratio from three immunoblot bands was quantified. (J, K) Knockdown of RAI14 in HEK293A (J) and SNU638 (K) cells increases the phosphorylation of LATS1 and YAP, whereas it decreases TAZ levels. The p-YAP/total-YAP, p-LATS1/total-LATS1 and TAZ/β-actin ratios from three immunoblot bands were quantified. (L, M) Knockdown of RAI14 in HEK293A (L) and AGS (M) cells reduces the nuclear localization of YAP. Scale bars: 10 μm. The quantification of the nuclear/cytoplasmic ratio and nuclear YAP in a single cell is shown. In (L), to quantify the YAP nuclear/cytoplasmic and nuclear YAP/DAPI ratios, 72, 74, and 78 cells were used for siCtrl, siRAI14#1, and siRAI14#2, respectively. In (M), 141, 119, and 122 cells were used for siCtrl, siRAI14#1, and siRAI14#2, respectively. (N) Phosphorylation of LATS1 and YAP is increased in perinatal gastric tissues of *Rai14* knockout mice, whereas the levels of TAZ are decreased. The p-YAP/total-YAP, p-LATS1/total-LATS1 and TAZ/β-actin ratios from three immunoblot bands were quantified. Data information: In (B–H, J, N), the error bars indicate the standard deviation ( ± s.d.) of triplicate measurements ((B, D, E, H, J, N) are biological replicates, and (C, F, G) are technical replicates). In (L, M), the error bars indicate the standard error of the mean ( ± s.e.m.) of the measurements (technical replicates). Statistical analysis was performed using two-tailed unpaired *t* test, and exact *P* values are shown in each figure; *P* < 0.05, statistically significant. Black or colored dots on the graphs indicate each measurement. Source data are available online for this figure.

RAI14 levels are positively correlated with the expression of YAP target genes. RAI14 interacts with NF2 and the interaction occurs at F-actin. Low mechanical forces or depletion of RAI14 induce dissociation of NF2 from F-actin, leading to increased interaction between NF2 and LATS1, phosphorylation of LATS1 and activation of the Hippo signaling pathway. Analysis of the Asian Cancer Research Group (ACRG) gastric cancer (GC) database and tissue from GC patients revealed higher expression of RAI14 in the diffuse type of GC, which has an ECM of high stiffness. Taken together, we report that RAI14 links mechanical forces induced by F-actin assembly to Hippo signaling, primarily via NF2, and suggest that increased RAI14 may promote the Hippo-YAP-dependent diffuse type of GC progression.

## Results

### RAI14 activates YAP/TAZ

To identify novel regulators of Hippo signaling, we used The Cancer Genome Atlas (TCGA) database, which provides comprehensive gene expression profiling and survival data of cancer patients (Weinstein et al, 2013). First, we listed the top fifteen differentially expressed genes associated with poor prognosis when expressed at high levels in stomach adenocarcinoma (STAD) (Fig. EV1A). Since these genes may accelerate GC progression, and since high levels of the Hippo signaling effector YAP are implicated in cancer, we examined the correlation between the expression of these genes and YAP transcriptional output. We also looked at which of the genes on the list were up-regulated in GC tissue. As a result, *RAI14* and *SERPINE1* were identified as meeting both of the above criteria (Fig. 1A). As the relationship between YAP and *SERPINE1* has already been studied (Marquard et al, 2020), we focused on investigating *RAI14* as a potential regulator of Hippo signaling and YAP activity. As described above, high expression of *RAI14* in GC patients was associated with poor prognosis (Fig. EV1B), and *RAI14* was highly expressed in gastric tumors compared to normal tissue (Fig. EV1C). Furthermore, the expression of *RAI14* and representative YAP target genes such as *CTGF* and *CYR61* were positively correlated in GC tissues (Fig. EV1D).

Based on the analysis of the TCGA database, we hypothesized that RAI14 may potentiate YAP transcriptional activity. Overexpression of RAI14 increased YAP reporter activity in HEK293A cell, GC cell lines (Fig. 1B) and several cell lines (Appendix

Fig. S1A,B) and the expression of several YAP target genes such as *CTGF*, *CYR61* and *ANKRD1* (Fig. 1C). On the contrary, depletion of RAI14 in HEK293A cells by siRNA or CRISPR-Cas9-mediated method resulted in a reduction of YAP reporter activity (Fig. 1D,E) as well as the expression of YAP target genes (Fig. 1F,G). Since the transcriptional activity of YAP is strongly linked to its phosphorylation state and cellular localization, we tested whether modulation of RAI14 levels also influences these states. Overexpression of RAI14 in HEK293A and AGS cells reduced the phosphorylation of YAP but increased the protein levels of YAP and its paralog TAZ (Fig. 1H,I). On the contrary, RAI14 knockdown induced the phosphorylation of LATS1 and YAP, the destabilization of TAZ (Fig. 1J,K). Furthermore, immunofluorescence assays demonstrated that RAI14 knockdown reduced the nuclear/cytoplasmic ratio of YAP (Fig. 1L,M). Since *Rai14* knockout results in perinatal lethality in mice (Kim et al, 2022), we analyzed the level of activation of Hippo signaling in the gastric tissue of *Rai14* knockout mice still alive in the perinatal state. *Rai14* knockout resulted in upregulation of LATS1 and YAP phosphorylation and destabilization of TAZ (Fig. 1N). These data suggest that RAI14 deficiency leads to activation of the Hippo signaling pathway with phosphorylation of LATS1, thereby promoting phosphorylation of YAP and inhibiting its transcriptional activity.

### RAI14 interacts with NF2

Since RAI14 deficiency led to activation of the Hippo signaling pathway, we investigated which Hippo signaling components interact with RAI14. Co-immunoprecipitation assay revealed that overexpressed RAI14 did not interact with YAP, TAZ, TEAD1, LATS1 and MST1/2 (Appendix Fig. S2A–E) but did interact with NF2 (Fig. 2A). Furthermore, RAI14 and NF2 were shown to interact at the endogenous level (Fig. 2B). Proximity-ligation assay (PLA) is known to be a valuable tool for the spatial detection of protein-protein interactions in cells (Söderberg et al, 2006). Using PLA, we found that NF2 and RAI14 interacted with each other, and that the interaction was mainly detected in the cytoplasm in HEK293A cells and GC cell lines (Fig. 2C–E). We confirmed that the antibodies used to detect NF2 and RAI14 were specific for their respective target proteins (Appendix Fig. S2F,G). NF2 has an N-terminal FERM domain region followed by alpha-helical and C-terminal domains, and RAI14 has an ankyrin repeat domain and a coiled-coil domain (Fig. 2F). We found that

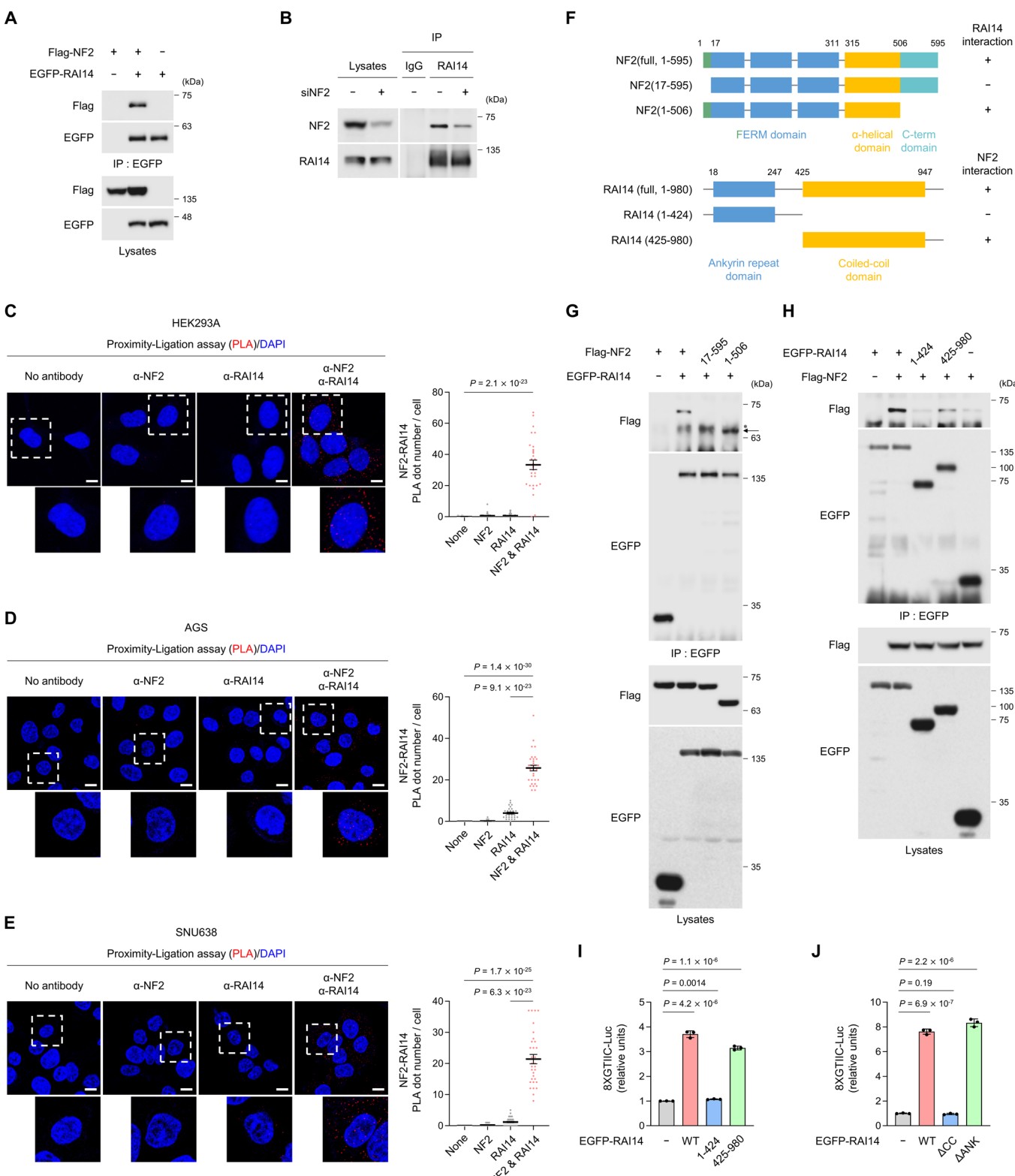

deletion of the N-terminal region of NF2 (17–595), but not the C-terminal region of NF2 (1–506), abolished the interaction with RAI14 (Fig. 2G). Furthermore, the ankyrin repeat domain-containing region of RAI14 (1–424) did not interact with NF2, whereas the coiled-coil region of RAI14 (425–980) did (Fig. 2H). These data suggest that the N-terminal region of NF2 and the coiled-coil region of RAI14 are responsible for their interaction.

◀ **Figure 2. RAI14 interacts with NF2.**

(A) Overexpressed Flag-NF2 in HEK293T cells interacts with EGFP-RAI14. (B) In HEK293A cells, endogenous NF2 interacts with RAI14. (C–E) Endogenous NF2 and RAI14 are in proximity. Scale bars: 10 μm. The quantification of the number of dots in a single cell is shown. In (C), 50, 31, 42, and 30 cells were used for none, NF2, RAI14, and NF2&RAI14, respectively. In (D), 38, 48, 33, and 32 cells were used. In (E), 41, 37, 39, and 32 cells were used. (F) Schematic diagrams for the wild-type and mutant forms of human NF2 and RAI14 and summary for their interaction. ( + ) indicates positive interaction. (G) The N-terminal region of NF2 is essential for the interaction with RAI14. HEK293T cells were used. (H) The coiled-coil domain region of RAI14 is essential for interaction with NF2. HEK293T cells were used. (I, J) Overexpression of the coiled-coil domain region of RAI14 in AGS cells activates YAP-reporter activity. Data information: In (C–E), the error bars indicate ± s.e.m. of the measurements (technical replicates). In (I, J), The error bars indicate ± s.d. of triplicate measurements (biological replicates). Statistical analysis was performed using two-tailed unpaired $t$ test, and exact $P$ values are shown in each figure; $P < 0.05$, statistically significant. Black or colored dots on the graphs indicate individual measurements. Source data are available online for this figure.

Based on these results, we tested whether the coiled-coil region of RAI14 is essential for the regulation of YAP transcriptional activity. We found that overexpression of the coiled-coil region of RAI14 (425–980), but not the ankyrin repeat domain region (1–424), upregulated YAP reporter activity (Fig. 2I). Similarly, RAI14 with the coiled-coil domain-truncated form did not upregulate YAP reporter activity, whereas the ankyrin repeat domain-truncated form did (Fig. 2J). These results indicate that the coiled-coil domain of RAI14 is essential for the regulation of YAP transcriptional activity.

## Regulation of YAP activity by RAI14 is mainly mediated through NF2

The above results suggest that RAI14 may regulate YAP transcriptional activity through NF2. Knockdown of RAI14 effectively induced YAP phosphorylation, whereas knockdown of both RAI14 and NF2 alleviated YAP phosphorylation (Fig. 3A). Similar to the phosphorylation status of YAP, knockdown of RAI14 induced the cytoplasmic localization of YAP, whereas knockdown of both RAI14 and NF2 maintained the nuclear localization of YAP (Fig. 3B). Furthermore, knockdown of both RAI14 and NF2 rescued the expression of YAP target genes compared to knockdown of RAI14 alone (Fig. 3C). However, we observed that loss of RAI14 still induced YAP phosphorylation (Fig. 3A). We think that this is because the remaining NF2 is still functioning or that there is an NF2-independent pathway for loss of RAI14 to induce phosphorylation of YAP. To resolve this concern, we used MDA-MB-231 cells, which have been reported to have extremely low expression of NF2 due to a homozygous non-sense mutation of NF2 (Dupont et al, 2011). In MDA-MB-231 cells, knockdown of RAI14 alone was ineffective in inducing YAP phosphorylation (Fig. 3D) or downregulating YAP target gene expression (Fig. 3E). However, RAI14 knockdown in combination with NF2 overexpression resulted in YAP phosphorylation and a reduction in YAP target gene expression in MDA-MB-231 cells (Fig. 3D,E). Although we cannot completely exclude the possibility that RAI14 depletion activates Hippo signaling in an NF2 independent manner, our data suggest that NF2 mainly mediates phosphorylation of YAP and downregulation of YAP target gene expression when the RAI14 levels are reduced.

## Destabilization of F-actin induces RAI14 proteasomal degradation

Previous studies have shown that RAI14 is associated with F-actin and can regulate F-actin integrity (Kim et al, 2022; Zhang et al, 2023). However, how F-actin integrity could regulate RAI14 was unknown. Furthermore, in our context, knockdown of RAI14 showed a marginal change in F-actin stability (Fig. EV2A). Since destabilization of F-actin is known to be an activating factor of Hippo signaling and an inhibitory factor of YAP activity, we wondered whether destabilization of F-actin would somehow affect RAI14, leading to activation of Hippo signaling and inhibition of YAP. Inhibition of actin polymerization in HEK293A cells using latrunculin A (Lat. A) showed increased phosphorylation of YAP and decreased RAI14 protein levels (Fig. 4A). Prolonged treatment with Lat. A further reduced RAI14 protein levels (Fig. 4B). It is well known that incubating cells in soft ECM conditions promotes F-actin destabilization and YAP inhibition (Dupont et al, 2011). Similar to Lat. A treatment, incubation of cells in soft ECM led to a reduction in RAI14 protein levels (Fig. 4C). Polymerization of F-actin requires Rho-associated protein kinase (ROCK) activity. Therefore, ROCK inhibitors are widely used to induce F-actin destabilization (Amano et al, 2010). We also found that inhibition of ROCK by treatment with Y-27632 reduced RAI14 protein levels (Fig. 4D). These data show that destabilization of F-actin induces a reduction in RAI14 protein levels, suggesting that mechanical forces applied to cells can modulate cellular RAI14.

We next asked how RAI14 protein levels are reduced when F-actin is destabilized. RAI14 mRNA levels were not affected by treatment with Lat. B, indicating that RAI14 protein stability was affected (Fig. EV2B). As the reduction in protein levels is mainly due to proteasomal or lysosomal degradation, we tested which degradation process affects RAI14 protein levels. Treatment with the proteasomal inhibitor MG132 blocked the reduction of RAI14 protein levels by Lat. A, whereas the lysosomal inhibitor bafilomycin A1 (Baf. A1) did not (Figs. 4E,F and EV2C). This suggests that the degradation of RAI14 upon F-actin destabilization is proteasome-dependent. However, the ubiquitination level of endogenous RAI14 was not increased upon F-actin destabilization (Fig. EV2D). Thus, proteasomal degradation of RAI14 appears to be ubiquitination-independent. Neddylation refers to the attachment of the small ubiquitin-like protein Nedd8 to target proteins (Enchev et al, 2015). It has been reported that neddylation can also lead to proteasomal degradation of target proteins. Treatment with MLN-4924, an inhibitor of neddylation (Soucy et al, 2009), efficiently blocked proteasomal degradation of RAI14 induced by Lat. B (Fig. 4G). We also found that the neddylation level of RAI14 was upregulated in the cells with the treatment of Lat. B (Fig. 4H). This suggests that RAI14 downregulation under F-actin destabilization depends on neddylation-mediated proteasomal degradation.

We also observed a mobility shift of RAI14 band at low mechanical force. We investigated whether the mobility shift was

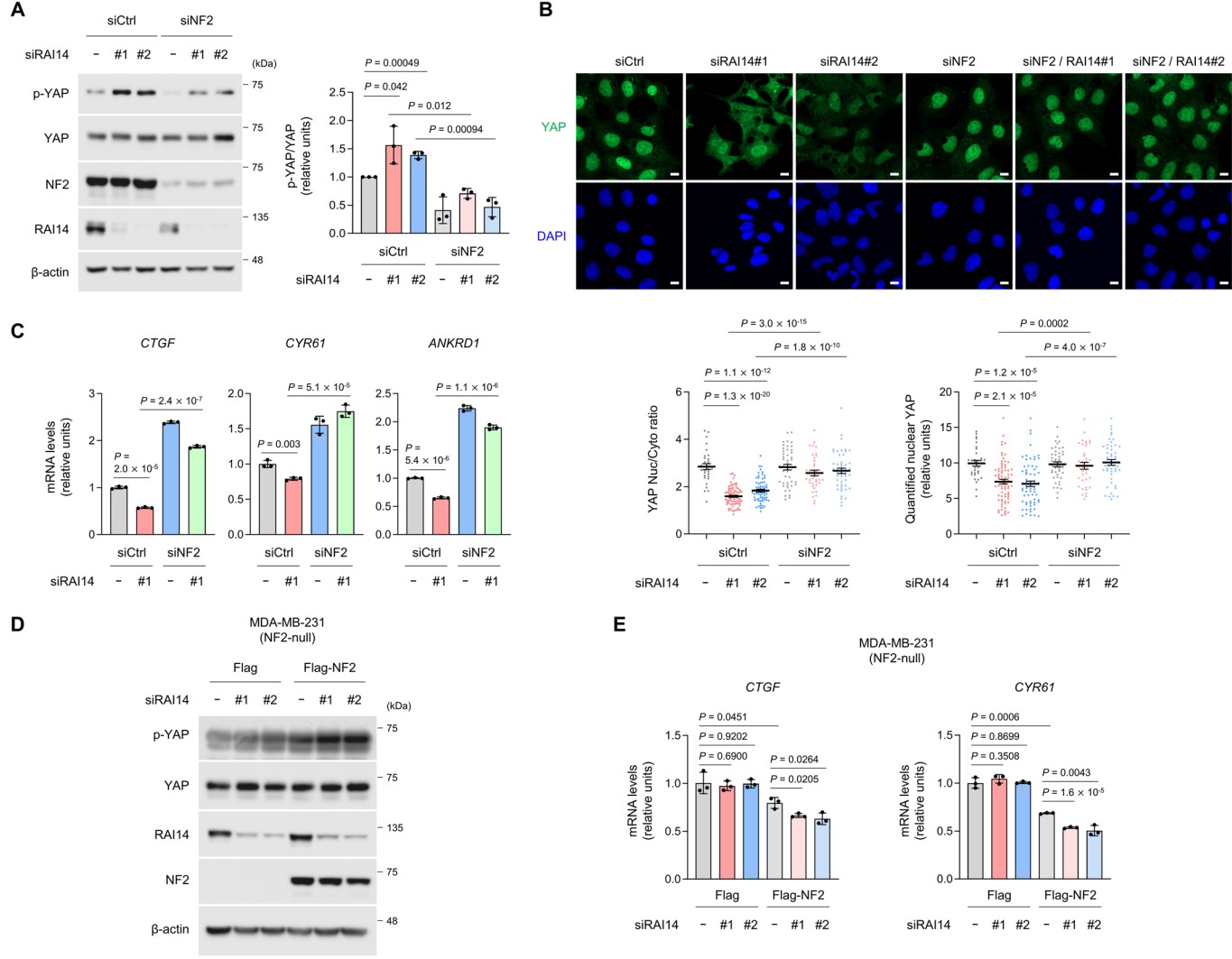

**Figure 3. NF2 mediates regulation of YAP/TAZ activity via RAI14.**

(A) NF2 is required for RAI14 knockdown-mediated YAP phosphorylation. The p-YAP/total-YAP ratio from three immunoblot bands was quantified. (B) NF2 is required for RAI14 knockdown-mediated cytoplasmic localization of YAP. Scale bars: 10 μm. Quantification of the nuclear/cytoplasmic ratio in a single cell is shown. To quantify the YAP nuclear/cytoplasmic and nuclear YAP/DAPI ratios, 31, 75, 67, 46, 37, and 50 cells were used for siCtrl, siRAI14#1, siRAI14#2, siNF2, siNF2/RAI14#1, and siNF2 / RAI14#2, respectively. (C) Downregulation of YAP-target gene expression by RAI14 knockdown is mediated by NF2. (D, E) NF2 is required for RAI14 knockdown-mediated YAP phosphorylation (D) and downregulation of the expression of YAP-target genes (E). Data information: In (A–C), HEK293A cells were used. In (D, E), MDA-MB-231 cells were used. In (A, C, E), the error bars indicate ± s.d. of triplicate measurements (A is biological replicates, and (C, E) are technical replicates). In (B), The error bars indicate ± s.e.m. of the measurements (technical replicates). Statistical analysis was performed using two-tailed unpaired t test, and exact P values are shown in each figure; P < 0.05, statistically significant. Black or colored dots on the graphs indicate each measurement. Source data are available online for this figure.

due to phosphorylation of RAI14. Treatment with lambda phosphatase (λ-PPase) reversed the mobility shift of the band (Fig. EV2E), suggesting that the mobility shift may be due to phosphorylation of RAI14, although we did not further investigate the mechanism and role of phosphorylation on RAI14.

Serum deprivation is known to activate Hippo signaling and inhibit the transcriptional activity of YAP (Yu et al, 2012). We tested whether serum deprivation could affect RAI14 protein levels in a similar way to F-actin destabilization and found that it reduced RAI14 protein levels (Fig. 4I). As previous studies have shown, serum deprivation affects the dynamics of F-actin integrity (Yu et al, 2012); we also confirmed that it reduced F-actin formation

(Fig. 4J). Our data suggest that factors that activate the Hippo signaling, such as serum deprivation, lead to the destabilization of F-actin and the reduction of RAI14 protein levels.

## RAI14 and NF2 interaction occurs on F-actin

One of the next primary questions was how RAI14 regulates Hippo signaling during changes in F-actin dynamics. We focused on NF2 because it is known that NF2 deficiency alleviates YAP phosphorylation during F-actin destabilization (Plouffe et al, 2016). A previous study proposing a parallel model of Hippo signaling suggested that NF2 interacts with LATS1/2 at the plasma

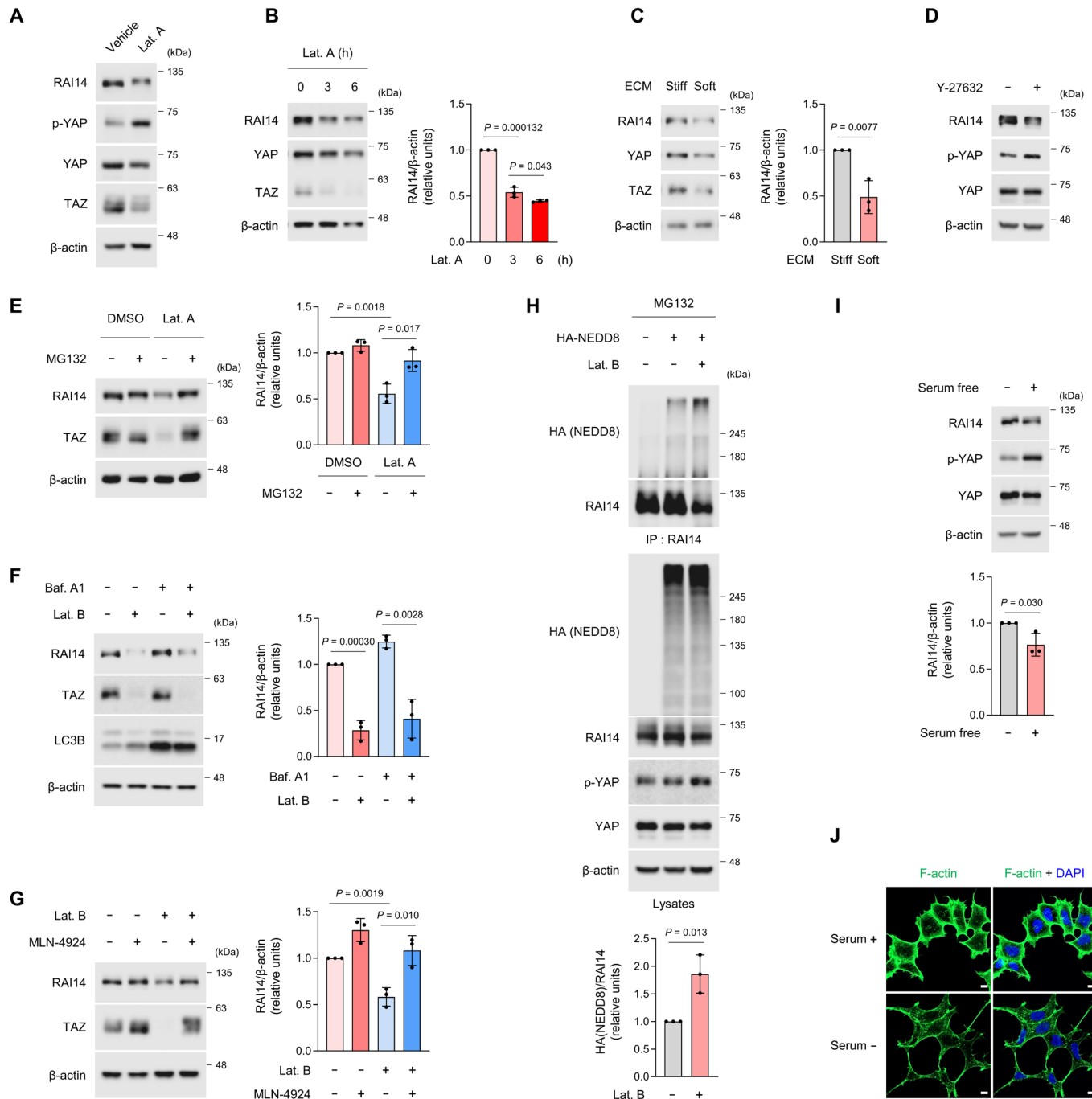

membrane, and the NF2-interacting LATS1/2 is phosphorylated by MST1/2 (Yin et al, 2013). Notably, it has been shown in *Drosophila* that destabilization of F-actin enhances the interaction between NF2 and Warts (LATS1/2 in mammals), suggesting that NF2 may be an F-actin-mediated regulator of Hippo signaling. However, how destabilization of F-actin promotes the interaction between NF2 and LATS1/2 remains to be elucidated.

We hypothesized that NF2 interacts with F-actin and that the released NF2, by destabilizing of F-actin, could interact with more LATS1. We expected RAI14 to be involved in this process. We then tested whether NF2 and RAI14 could interact with F-actin.

Immunofluorescence showed that some regions of stained NF2 and F-actin colocalize (Fig. 5A), and similar results were obtained for RAI14 and F-actin (Fig. EV3A). Colocalization of RAI14 and F-actin was also confirmed in mouse small intestine and gastric organoids (Fig. EV3B,C). The PLA also revealed the interaction between NF2 and β-actin, and treatment with Lat. B effectively reduced PLA signals (Fig. EV3D,E). This suggests that the interaction between NF2 and β-actin occurred primarily on F-actin, but not on globular actin (G-actin). A similar result was obtained using the PLA between RAI14 and β-actin (Fig. EV3F,G). The F-actin binding assay showed that treatment with Lat.

◄  **Figure 4.  Destabilization of F-actin induces proteasomal degradation of RAI14.**

(A, B) Destabilization of F-actin by Latrunculin A (Lat. A) reduces RAI14 protein levels. In (A), DMSO as vehicle or Lat. A (2 μM) for 6 h was used. In (B), the RAI14/β-actin ratio from three immunoblot bands was quantified. (C) Incubation of cells on a soft matrix (1 kPa) reduces RAI14 protein levels. HEK293A cells were incubated on a stiff (50 kPa) or soft (1 kPa) matrix for one day. The RAI14/β-actin ratio from three immunoblot bands was quantified. (D) Inhibition of F-actin polymerization by the Rho-associated kinases (ROCK) inhibitor Y-27632 reduces RAI14 protein levels. Y-27632 (50 μM) for 6 h was used. (E) RAI14 protein degradation via F-actin destabilization is mediated by the proteasomal degradation pathway. Lat. A (2 μM) and the proteasome inhibitor MG132 (25 μM) for 6 h were used as indicated in the figure. The RAI14/β-actin ratio from three immunoblot bands was quantified. (F) The lysosomal degradation pathway is not responsible for RAI14 protein degradation via F-actin destabilization. Lat. A (2 μM) and the lysosome inhibitor Bafilomycin A1 (Baf. A1, 100 nM) for 6 h were used as indicated in the figure. The RAI14/β-actin ratio from three immunoblot bands was quantified. (G) Proteasomal degradation of RAI14 protein via F-actin destabilization is dependent on neddylation. Lat. B (2 μM) and the Nedd8 activating enzyme inhibitor MLN4924 (750 nM) for 6 h were used as indicated in the figure. The RAI14/β-actin ratio from three immunoblot bands was quantified. (H) F-actin destabilization increases the neddylation level of RAI14. HEK293T cells were transfected with HA-NEDD8 as shown in the figure and treated with Lat. B (2 μM) and MG132 (25 μM) for 6 h. The HA(NEDD8)/RAI14 ratio from three immunoblot bands was quantified. (I) Serum deprivation reduces RAI14 protein levels. Serum deprivation for 6 h was performed. The RAI14/β-actin ratio from three immunoblot bands was quantified. (J) Serum deprivation reduced F-actin formation. Serum deprivation for 6 h was performed. For F-actin staining, phalloidin was used. Scale bars: 10 μm. Data information: HEK293A cells were used in all Figures except (H). In (B, C, E–I), the error bars indicate ± s.d. of triplicate measurements (biological replicates). Statistical analysis was performed using two-tailed unpaired *t* test, and exact *P* values are shown in each figure; *P* < 0.05, statistically significant. Black dots on the graphs indicate individual measurements. Source data are available online for this figure.

B decreased the interaction of RAI14 and NF2 with F-actin, suggesting that RAI14 and NF2 bind to F-actin (Fig. 5B). In addition, a coiled-coil domain-containing mutant form of RAI14, which interacts with NF2 and activates YAP transcriptional activity (Fig. 2H–J), interacted with F-actin (Fig. 5C). These results raise the possibility that the interaction between NF2 and RAI14 occurs on F-actin. Consistent with this possibility, PLA signals between NF2 and RAI14 colocalized with F-actin staining areas (Fig. 5D). Interestingly, we found that RAI14 knockdown reduced the interaction between NF2 and F-actin (Figs. 5E and EV3H), suggesting that RAI14 depletion causes NF2 to dissociate from F-actin (Fig. 5F).

## The interaction between RAI14, NF2, and LATS is regulated by ECM stiffness and F-actin integrity

Since loss of RAI14 results in dissociation of NF2 from F-actin and activation of Hippo signaling, we investigated whether destabilization of F-actin or depletion of RAI14 reduces the interaction between RAI14 and NF2 while increasing the interaction between NF2 and LATS1 (Fig. 6A). Incubation of cells in soft ECM to apply low mechanical forces or treatment of Lat. B decreased the interaction between NF2 and RAI14 (Figs. 6B–E and EV4C, quantification for the ratio of NF2/RAI14 shown in Fig. 6B,C was presented in EV4A,B, respectively), suggesting that F-actin destabilization promotes the release of NF2 from RAI14. We further found that application of low mechanical forces using low stiffness ECM and destabilization of F-actin upregulated the interaction between NF2 and LATS1 (Fig. 6F–H, the quantification for the ratio of NF2/Myc-LATS1 shown in Fig. 6H was presented in Fig. EV4D), suggesting that the finding from the previous study in *Drosophila* could be applied to mammalian cells (Yin et al, 2013).

Since we showed that the level of RAI14 was reduced and the Hippo signaling was activated via increased interaction of NF2 and LATS1 in low stiffness conditions (Figs. 4C and 6F), we examined whether the expression of RAI14 can override the low mechanics to activate nuclear YAP. In both stiff and soft matrix conditions, the cells overexpressing wild-type RAI14 (marked with a white arrow; upper panel) increased the nuclear localization of YAP, whereas the cells overexpressing the RAI14 (1–424) form (which does not interact with NF2 (Fig. 2F), marked with a white arrow; lower panel) did not promote the nuclear localization of YAP

(Fig EV4E). This suggests that overexpression of RAI14 overrides the low mechanics to promote nuclear localization of YAP by binding to NF2. To further substantiate our claim, we examined whether wild-type RAI14 or RAI14 (1-424) could rescue the reduced transcriptional activity of the YAP reporter in the low stiffness condition induced by Lat.B treatment. We found that overexpression of RAI14 partially rescued reporter activity, but not in the form of RAI14 (1–424) (Fig EV4F). Since incubation of cells on low stiffness induces the suppression of YAP transcriptional activity through various pathways in a Hippo-dependent and -independent manner, we believe that this is the reason why partial, but not complete, rescue was observed by RAI14 overexpression. Taken together, our data support the idea that the overexpression of RAI14 under low stiffness conditions can induce nuclear localization of YAP and increase expression of YAP target genes.

We next tried to investigate whether the interaction of RAI14 with F-actin is a crucial determinant in the regulation of the NF2-LATS complex formation, or whether RAI14 itself is a critical factor in this process. Since we showed that serum stimulation significantly elevated F-actin polymerization levels, we proceeded to investigate whether serum stimulation influenced the interaction between RAI14, NF2, and LATS1. Upon serum stimulation, we observed that the interaction between RAI14 and NF2 was increased, while the interaction between NF2 and LATS1 was decreased (Fig. 6I,J). This indicates that F-actin may facilitate the formation of the RAI14-NF2 complex. Next, we asked whether RAI14 alone could regulate the NF2-LATS interaction. RAI14 knockdown resulted in an increase in the interaction between NF2 and LATS1, as well as the phosphorylation of LATS1 and YAP (Fig. 6K, the quantification for the ratio of NF2/Myc-LATS1 was presented in EV4G). Furthermore, under conditions of F-actin destabilization, the knockdown of RAI14 resulted in a further increase in the interaction between NF2 and LATS1, as well as the phosphorylation of LATS1 and YAP (Fig. 6L). Conversely, RAI14 overexpression resulted in a reduction in the NF2-LATS1 interaction (Fig. 6M). It is noteworthy that, in the context of F-actin deficiency induced by Lat.B treatment, RAI14 overexpression was still able to reduce NF2-LATS1 interaction (Fig. 6N). This suggests that RAI14 may be capable of regulating the interaction between NF2 and LATS1 independently of F-actin. To corroborate the role of RAI14 in the absence of F-actin, we used the NF2

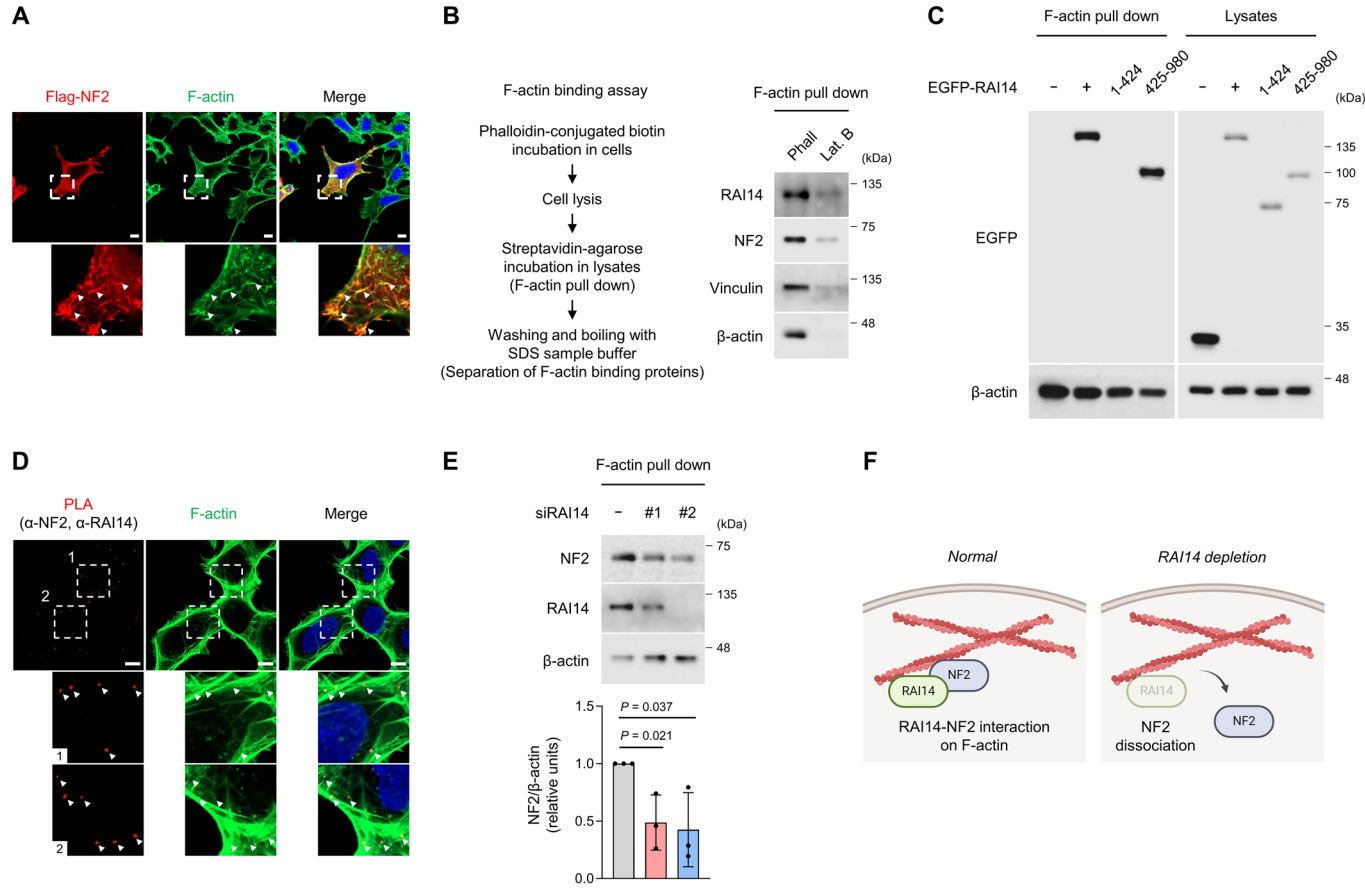

**Figure 5. RAI14 and NF2 interaction occurs on F-actin.**

(A) Overexpressed Flag-NF2 colocalizes with F-actin. For F-actin staining, phalloidin was used. Regions of NF2 and F-actin colocalization are marked with white arrowheads in the magnified regions. Scale bars: 10 μm. (B) NF2 and RAI14 interact with F-actin. Vinculin was used as a positive control for F-actin binding. (C) The coiled-coil domain region of RAI14 is essential for interaction with F-actin. (D) The interaction between RAI14 and NF2 on F-actin. PLA signals colocalizing with F-actin are marked with white arrowheads in the magnified regions 1 and 2. Scale bars: 10 μm. (E) Knockdown of RAI14 reduces the interaction between NF2 and F-actin. The NF2/β-actin ratio from three immunoblot bands was quantified. (F) Schematic representation of the release of NF2 from F-actin by RAI14 depletion. Data information: HEK293A cells were used in all Figures. In (E), Error bars indicate ± s.d. of triplicate measurements (biological replicates). Statistical analysis was performed using two-tailed unpaired $t$ test, and exact $P$ values are shown in each figure; $P < 0.05$, statistically significant. Black dots on the graph indicate individual measurements. Source data are available online for this figure.

(17–595) form, which does not interact with RAI14 (Fig. 2F). By performing the F-actin binding assay, we found that the NF2 (17–595) form had significantly reduced binding to F-actin compared to wild-type NF2 (Fig. EV4H). In addition, the NF2 (17–595) form showed increased interaction with LATS1 and phosphorylation of LATS1 compared to wild-type NF2, indicating that this mutant form has a greater ability to activate Hippo signaling (Fig. EV4I). Overall, these data suggest that NF2-LATS complex formation is inhibited by binding of RAI14 to NF2, and this inhibition is promoted by NF2 sequestration via RAI14 to F-actin.

## RAI14 promotes cell proliferation and growth in Hippo signaling-dependent manner

Since Hippo signaling is an important regulator of cell proliferation and growth, and RAI14 acts as a regulator of Hippo signaling, we hypothesized that RAI14 could regulate cell proliferation and

growth. Overexpression of RAI14 increased the growth rate and colony-forming ability of HEK293A, AGS, and SNU638 cells (Fig. 7A–F). However, treatment with verteporfin (VP), which inhibits the interaction between YAP and TEAD (Liu-Chittenden et al, 2012), reduced the colony-forming ability and cell growth rate induced by RAI14 overexpression (Fig. 7A,B). Given that VP is known to have side effects, an alternative TEAD inhibitor, MGH-CP1, which inhibits TEAD palmitoylation (Li et al, 2020), was employed. The results demonstrated that the upregulated cell growth rate and colony-forming capacity in AGS, and SNU638 cells were effectively blocked by treatment with MGH-CP1 (Fig. 7C–F). These data provide further support for the conclusion that RAI14 promotes cell proliferation and growth by regulating Hippo signaling. As with the effects of MGH-CP1, the effects of RAI14 overexpression were blocked by YAP/TAZ knockdown in HEK293A, AGS, and SNU638 cells (Appendix Fig. 3). Next, to determine the role of endogenous RAI14 in cancer cells, we generated stably expressing shRAI14 in SNU638 GC cells.

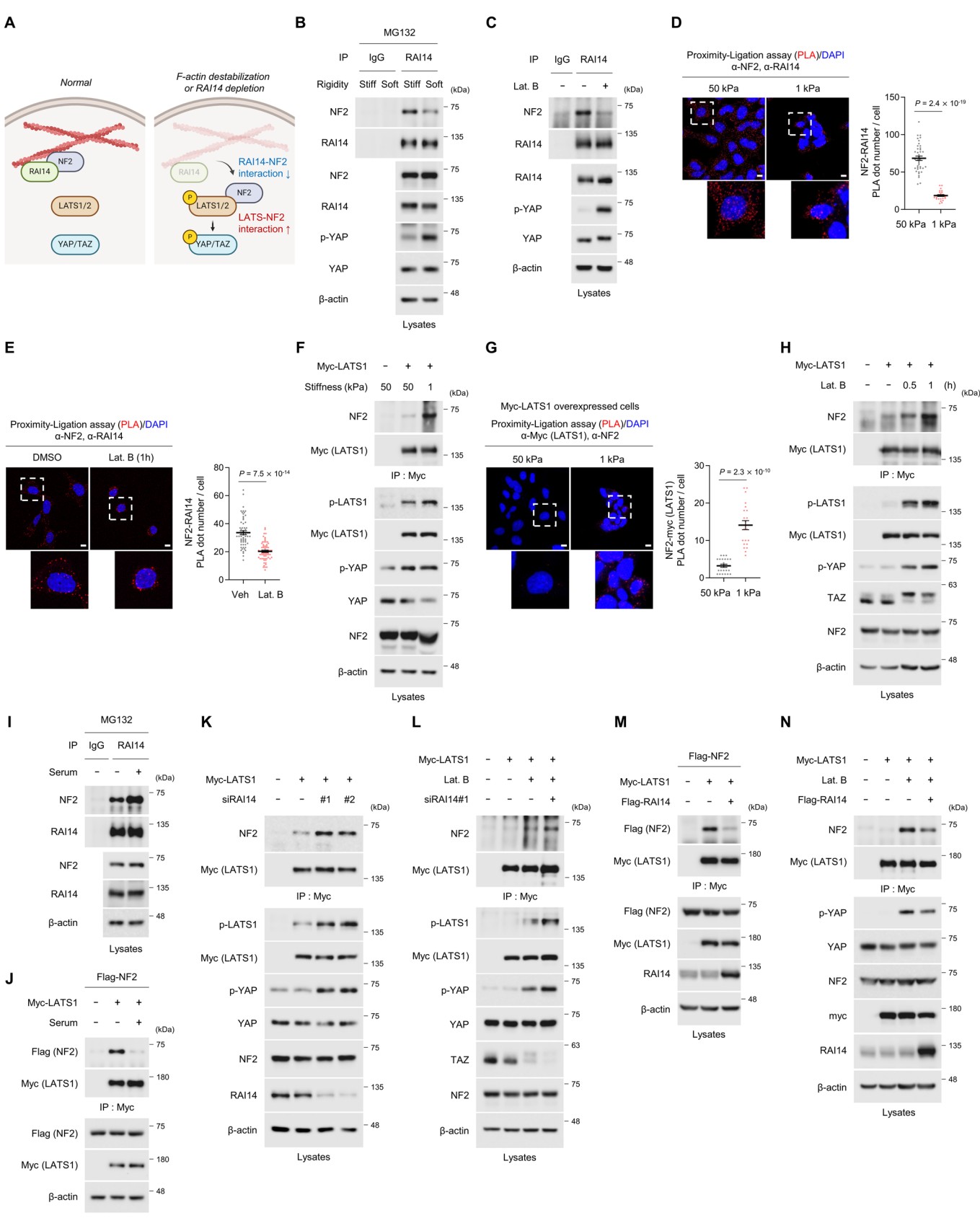

◀ **Figure 6. ECM stiffness and F-actin integrity regulate the interaction between RAI14, NF2 and LATS.**

(A) Schematic representation of how released NF2 could activate the Hippo signaling pathway. (B) Incubation of cells on the soft ECM results in reduced interaction between endogenous RAI14 and NF2. HEK293A cells were incubated on a stiff (64 kPa) or soft (0.5 kPa) matrix for one day and treated with the proteasomal inhibitor MG132 (25 μM) for 6 h. The immunoprecipitated NF2/RAI14 ratio from three immunoblot bands was quantified in Fig. EV4A. (C) Destabilization of F-actin reduces the interaction between endogenous NF2 and RAI14. HEK293A cells were treated with Lat. B for 1 h. The immunoprecipitated NF2/RAI14 ratio from three immunoblot bands was quantified in Fig. EV4B. (D, E) PLA assay showing that the interaction between NF2 and RAI14 is inhibited by incubation on soft ECM or Lat. B treatment. HEK293A cells were incubated on soft ECM for one day (D) or treated with Lat. B for 1 h (E). In (D), MG132 (25 μM) was treated for 6 h prior to the PLA assay. The quantification of the number of dots in a single cell is shown in the right panel. In (D), 40 and 26 cells were used for 50 kPa and 1 kPa, respectively. In (E), 59 and 66 cells were used for DMSO and Lat. B, respectively. Scale bars: 10 μm. (F) Incubation of cells on the soft ECM results in the upregulation of NF2 and Myc-LATS1 interaction. After transfection, HEK293A cells were incubated on a stiff (50 kPa) or soft (1 kPa) matrix for one day. (G) PLA assay shows that the interaction between NF2 and Myc-LATS1 is increased upon incubation on soft ECM. After transfection, HEK293A cells were incubated on stiff (50 kPa) or soft (1 kPa) matrix for one day. Quantification of the number of dots in a single cell is shown in the right panel. 21 and 22 cells were used for 50 kPa and 1 kPa, respectively. Scale bars: 10 μm. (H) Destabilization of F-actin increases the interaction between Myc-LATS1 and NF2. In HEK293A cells, Lat. B was treated one day after transfection, as indicated in the figure. The immunoprecipitated NF2/Myc(LATS1) ratio from three immunoblot bands was quantified in Fig. EV4C. (I) Serum stimulation increases the interaction between RAI14 and NF2. HEK293A cells were treated with serum (10% FBS) and MG132 (25 μM) for 6 h. (J) Serum stimulation reduces the interaction between Flag-NF2 and Myc-LATS1. HEK293A cells were transfected as indicated in the figure and treated with serum (10% FBS) for 6 h. (K) Knockdown of RAI14 increases the interaction between Myc-LATS1 and NF2. HEK293A cells were transfected as shown in the figure. The immunoprecipitated NF2/Myc(LATS1) ratio from three immunoblot bands was quantified in Fig. EV4G. (L) Destabilization of F-actin with knockdown of RAI14 synergistically increases the interaction between Myc-LATS1 and NF2. HEK293A cells were transfected as indicated in the figure and treated with Lat. B for 1 h. (M) Overexpression of Flag-RAI14 reduces the interaction between Flag-NF2 and Myc-LATS1. HEK293A cells were used. (N) Overexpression of Flag-RAI14 partially rescues the interaction between NF2 and Myc-LATS1, which is upregulated by F-actin destabilization. HEK293A cells were transfected as indicated in the figure and treated with Lat. B for 1 h. Data information: In (D, E, G), the error bars indicate ± s.e.m. of the measurements (technical replicates). Statistical analysis was performed using two-tailed unpaired t test, and exact P values are shown in each figure; P < 0.05, statistically significant. Colored dots on the graphs indicate individual measurements. Source data are available online for this figure.

shRAI14-expressing cells showed reduced YAP transcriptional activity (Fig. EV5A), nuclear localization of YAP (Fig. EV5B), cell growth rate (Fig. 7G) and colony-forming ability (Fig. 7H). To confirm that these effects were due to activation of Hippo signaling, we used TRULI, which blocks Hippo signaling by inhibiting LATS1/2 (Kastan et al, 2021). Treatment with TRULI significantly, but not completely, rescued the shRAI14-mediated reduction in colony-forming ability and cell growth rate (Fig. 7G,H), suggesting that Hippo signaling is involved in RAI14-mediated cell proliferation and growth regulation, although we cannot exclude the possibility that there are additional Hippo-independent roles of RAI14 in regulating cell proliferation and growth regulation.

## RAI14-Hippo signaling is implicated in GC progression

Previous studies have suggested that RAI14 is involved in GC progression (Zhou et al, 2015), but the detailed molecular mechanism is not well understood. Since our data suggested that RAI14 transmits mechanical forces to Hippo signaling, we wondered whether this pathway was involved in GC progression. According to Lauren's classification in 1965, GC can be divided into two histological types: intestinal and diffuse-type (Lauren, 1965). Intestinal-type GC (IGC) is characterized by tubular gland-like structures, relatively well-differentiated histology and a better prognosis (Adachi et al, 2000; Ribeiro et al, 1981). Meanwhile, diffuse-type GC (DGC) is characterized by poorly cohesive cancer cells, highly invasive tumor character, relatively poorly differentiated histology and a worse prognosis (Adachi et al, 2000; Ribeiro et al, 1981). In addition, it has been reported that DGC tissues form a remodeled ECM structure that favors cancer progression (Yang et al, 2021), and cells with DGC characteristics have activated YAP and F-actin formation (Zhang et al, 2020). Therefore, we hypothesized that RAI14-Hippo signaling is more involved in DGC than in IGC.

Next, we focused on analyzing the GC database from the ACRG, which proposed a molecular classification of GC subtypes based on a large sample size (300 cases) and data from long-term follow-up of all patients (Cristescu et al, 2015). Analyzing the ACRG GC database, we found that RAI14 expression was upregulated in both intestinal and diffuse-type gastric tumors compared to adjacent normal tissue. Interestingly, RAI14 levels were significantly higher in diffuse-type tumors than in intestinal-type tumors (Fig. EV5C). Survival analyses showed that high RAI14 expression was significantly associated with worse prognosis in DGC, but its prognostic significance was not identified in IGC (Fig. EV5D). In addition, there was a positive correlation between the expression of RAI14 and representative YAP target genes in DGC patients (Fig. EV5E), consistent with the results derived from the TCGA analysis. Next, we validated our results using clinical tissue samples from GC patients (8 IGC and 8 DGC) and found that the protein level of RAI14 was upregulated in most tumor tissues regardless of tumor histology. Interestingly, however, the pattern of increase was more pronounced in diffuse-type tumors, and the protein levels of YAP, TAZ, F-actin, and CTGF were also significantly upregulated, similar to the RAI14 pattern (Figs. 7I–K and EV5F). Consistent with previous reports, these data suggest that high mechanical forces due to increased tissue stiffness lead to upregulation of RAI14 and enhancement of YAP/TAZ activity in DGC tissues (Jang et al, 2021; Yang et al, 2021).

In addition to the Lauren classification, a previous report from the ACRG GC database proposed that GC can be classified into four groups based on molecular subtypes (Cristescu et al, 2015). Microsatellite instable (MSI)-type tumors are similar to the intestinal-type and have the best prognosis, whereas microsatellite stable with epithelial-to-mesenchymal transition (MSS/EMT)-type tumors show diffuse-type features and have the worst prognosis. The others are classified by TP53 activity, with MSS/TP53-active (MSS/TP53$^+$) and MSS/TP53-inactive (MSS/TP53$^-$) tumors showing intermediate prognosis (Fig. EV5G). We found that CTGF, CYR61 and RAI14 gene expression was highest in the MSS/EMT

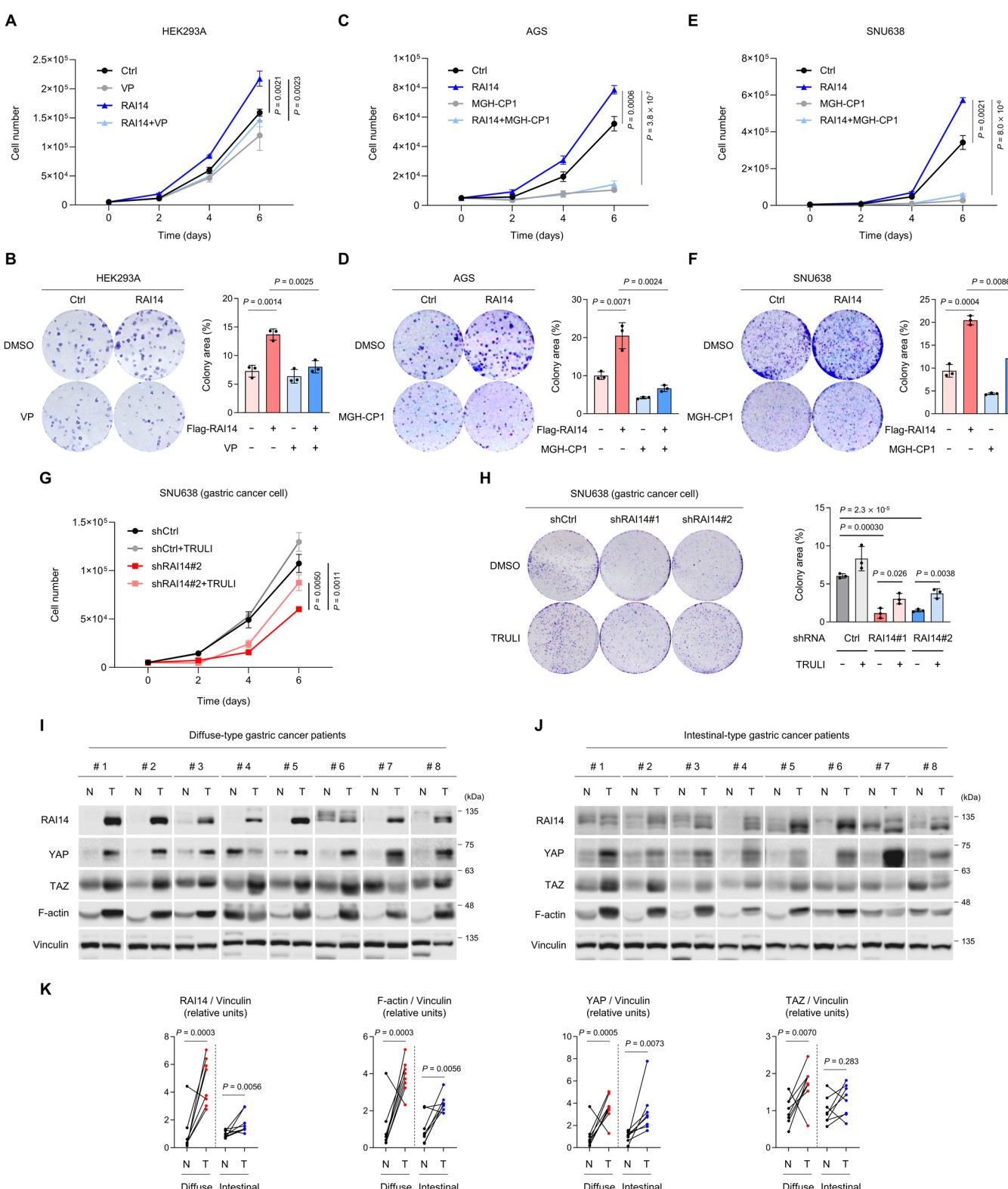

type, suggesting that three genes are consistently upregulated in gastric cancers with diffuse-type feature regardless of tumor classification (Fig. EV5H–J). 82% of MSS/EMT-type tumors are diffuse type with high expression of RAI14 (Fig. EV5G), and these patients expressing high RAI14 had a much worse prognosis than MSS/EMT-type patients expressing low RAI14 (Fig. EV5K). Taken together, our experimental data with human GC tissues and analysis of the ACRG and TCGA databases suggest that the RAI14-Hippo signaling axis plays a pivotal role in GC progression and that RAI14 may be a robust biomarker for DGC.

◀ **Figure 7. RAI14-Hippo signaling promotes cell proliferation and growth and is involved in gastric cancer progression.**

(A, B) Overexpressed RAI14 increases cell proliferation and colony-forming ability, but treatment with verteporfin (VP) attenuates the effect of RAI14 overexpression. HEK293A cells were treated with DMSO or verteporfin (0.5 µg/ml) as indicated in the figure. For (B), the quantification of "colony area per area of the well" is shown. (C–F) Overexpressed RAI14 increases cell proliferation and colony-forming ability, but treatment with MGH-CP1 attenuates the effect of RAI14 overexpression. AGS (C, D) and SNU638 (E, F) cells were treated with MGH-CP1 (10 µM in (C, E), 3 µM in (D, F)) as indicated in the figure. For (D, F), the quantification of "colony area per area of the well" is shown. (G, H) Knockdown of RAI14 reduces cell proliferation and colony-forming ability, but treatment with TRULI alleviates the effect of RAI14 knockdown. shRNA-containing lentivirus-transduced SNU638 cells were treated with TRULI (10 µM) as indicated in the figure. For (H), the quantification of "colony area per area of the well" is shown. (I, J) RAI14, YAP, TAZ and F-actin levels are high in GC tissues and significantly higher in DGC. (K) Quantification of RAI14/Vinculin, F-actin/Vinculin, YAP/Vinculin, and TAZ/Vinculin ratios from immunoblots (I, J). Quantified values of tumor and normal tissue samples are indicated by black (normal), red (diffuse type tumor) and blue (intestinal type tumor) dots. Black lines connect tumor and normal tissues from the same patient. Data information: In (A–H), the error bars indicate ± s.d. of triplicate measurements (biological replicates). Statistical analysis was performed using two-tailed unpaired *t* test (A–H, K), and exact *P* values are shown in each figure; $P < 0.05$, statistically significant. Black dots on the graphs indicate individual measurements. Source data are available online for this figure.

## Discussion

Although the relationship between mechanotransduction and Hippo signaling was reported about a decade ago, how F-actin, which is increased by mechanical forces, regulates Hippo signaling remains an open question. Here, we propose that RAI14 may be a regulator of Hippo signaling and a responder to mechanical forces. When cells are exposed to high mechanical forces, F-actin formation is promoted and the interaction between RAI14 and NF2 on F-actin is increased. In this state, the activity of LATS is kept low, leading to the activation of YAP/TAZ. Conversely, low mechanical forces destabilize F-actin, promoting the dissociation of NF2 from RAI14 and upregulating the interaction between LATS and NF2, thereby activating the Hippo signaling pathway (Fig. 8).

Although it is generally accepted that YAP/TAZ is involved in mechanotransduction, there has been controversy as to whether the process depends on Hippo signaling. The first study to report the relationship between YAP/TAZ and mechanotransduction suggested that Rho GTPase activity and actin cytoskeletal tension are essential for YAP/TAZ activity, but it is independent of Hippo signaling (Dupont et al, 2011). Another study showed that mechanical forces applied to cells lead to nuclear flattening, opening nuclear pores and facilitating nuclear translocation of YAP (Elosegui-Artola et al, 2017). The other study reported that mechanical forces induce the assembly of F-actin in the nucleus, thereby sequestering the switch/sucrose non-fermentable (SWI/SNF) chromatin remodeling complex on F-actin and enhancing the interaction between YAP and TEAD (Chang et al, 2018). These reports suggest that the regulation of YAP by mechanical forces is Hippo-independent. However, other studies have shown that YAP-mediated mechanotransduction relies on Hippo signaling. Modulation of F-actin assembly by inducing cell detachment, altering cell attachment regions or regulating actin-capping proteins affected the phosphorylation status of YAP and its transcriptional activity dependent on the Hippo kinases LATS1/2 (Fernández et al, 2011; Sansores-Garcia et al, 2011; Wada et al, 2011; Yu et al, 2012). It has been shown that the Ras-related GTPase RAP2 protein is activated by soft ECM and sequentially activates MAP4Ks and LATS1/2, resulting in phosphorylation of YAP and inhibition of its activity (Meng et al, 2018). Collectively, mechanical forces may regulate YAP activity in a Hippo-dependent or -independent manner in different cellular contexts. Our study suggests that RAI14 responds to mechanical force and regulates YAP in a Hippo-dependent manner.

RAI14 was first identified as a gene induced by retinoic acid treatment in retinal cells (Kutty et al, 2001). The biological role of RAI14 is primarily related to cancer progression and neuronal function. High expression of RAI14 is associated with poor prognosis in GC patients, and knockdown of RAI14 reduces GC cell proliferation (Zhou et al, 2015). Recent studies have shown that the deubiquitinase STAMBP suppresses the ubiquitination of RAI14 and stabilizes its protein level, thereby promoting breast cancer progression (Yang et al, 2022). However, little is known about how RAI14 promotes cancer progression and the signaling pathways involved in this process. Our data show that the cancer-promotive function of RAI14 is strongly associated with Hippo signaling and YAP activity. The high proliferative activity of RAI14-overexpressing cells was alleviated by pharmacological inhibition of YAP, and conversely, the low proliferation of RAI14-depleted cells was rescued by increasing YAP activity. In addition to cancer progression, RAI14 promotes neuronal function by regulating dendritic arborization and neuronal morphogenesis (Wolf et al, 2019). In addition, a recent study showed that *Rai14* knockout mice had reduced synaptic activity and exhibited depressive-like phenotypes (Kim et al, 2022). Several studies have reported downregulation of YAP activity in Alzheimer's disease (AD) patients, and overexpression of YAP improved neuronal function and cognitive performance in an AD mouse model (Tanaka et al, 2020; Xu et al, 2018). It will be interesting to see if our proposed model (Fig. 8) can also be applied to the neuronal function of RAI14.

NF2, also known as Merlin, acts as a tumor suppressor and mutations in the *NF2* gene lead to neurofibromatosis type 2 disease, which is characterized by the occurrence of bilateral vestibular schwannomas and other central nervous system tumors (Rouleau et al, 1993; Trofatter et al, 1993). The role of NF2 as a regulator of Hippo signaling was first identified in *Drosophila*. This study suggested that it acts upstream of Hippo (MST1/2 in mammals) and Warts (LATS1/2 in mammals) (Hamaratoglu et al, 2006). In addition to *Drosophila*, liver-specific conditional knockout of *NF2* in mice resulted in increased liver size and a hepatocellular carcinoma phenotype due to inactivation of Hippo signaling and hyperactivation of YAP (Zhang et al, 2010). Further studies showed that plasma membrane-localized NF2 interacts with LATS1/2 and facilitates the phosphorylation of LATS1/2 by MST1/2 (Yin et al, 2013). The authors showed that depolymerization of F-actin induces interaction between NF2 and LATS1/2, but the mechanism for the increased interaction is unclear. NF2 has also been reported to interact with F-actin, but

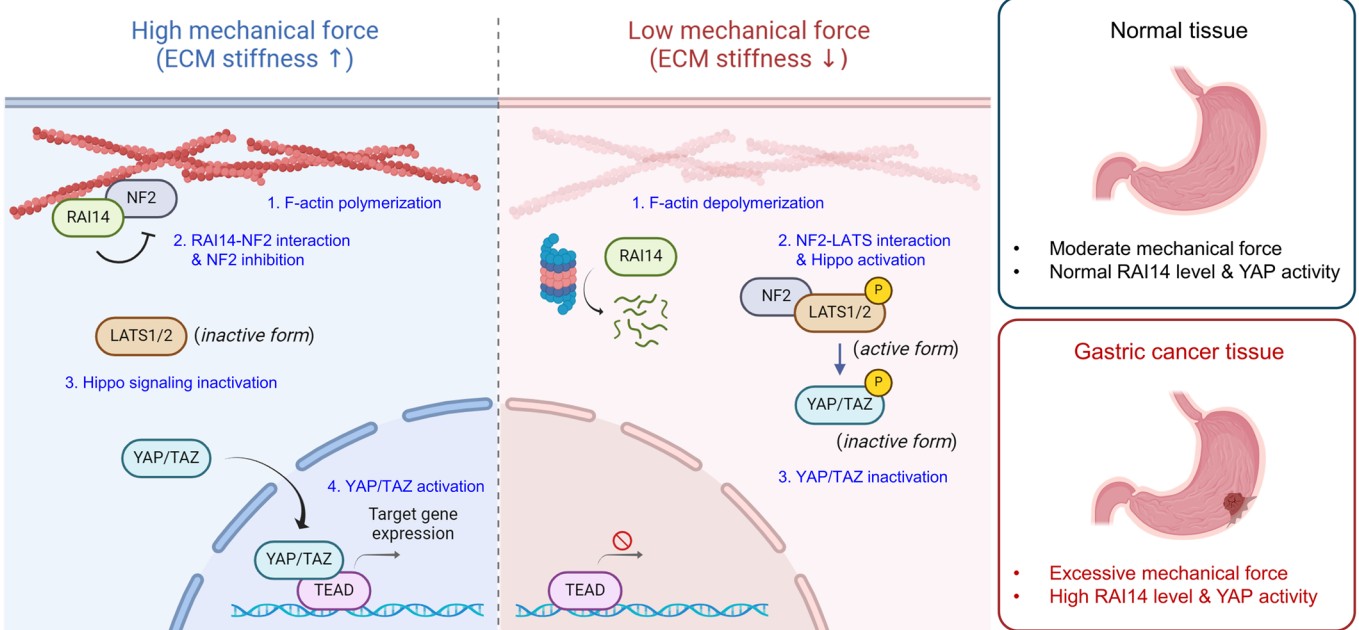

**Figure 8. Graphical summary of the RAI14-Hippo signaling pathway and its relationship with mechanical force and gastric cancer progression.**

When cells are exposed to high mechanical forces, such as when cells are grown on high stiffness ECM, polymerization of F-actin is promoted. In this condition, the interaction between RAI14 and NF2 on F-actin occurs and the function of NF2, known as an activator of the Hippo signaling pathway, is inhibited. LATS1/2 therefore remains in its inactive form, leading to an increase in unphosphorylated YAP/TAZ and nuclear localization and expression of their target genes. Conversely, under low mechanical forces, such as when cells are grown on low stiffness ECM, polymerization of F-actin is inhibited. NF2, released from F-actin and RAI14, then interacts with LATS1. This leads to the activation of the Hippo signaling pathway and sequential phosphorylation of LATS1/2 and YAP/TAZ, resulting in inactivation of YAP/TAZ. In GC tissue, the levels of tissue stiffness, RAI14 and YAP are high. Our model suggests that excessive mechanical force and high levels of RAI14 and YAP activity are involved in GC progression.

the role of F-actin-localized NF2 in regulating Hippo signaling is unknown (Brault et al, 2001; James et al, 2001). Our study shows that NF2 and RAI14 interact to be localized on F-actin. Loss of RAI14, low mechanical force or depolymerization of F-actin therefore releases NF2 from F-actin, and the released NF2 interacts with LATS1, leading to activation of the Hippo signaling pathway.

The diverse cellular localization of NF2 and its associated function have been investigated. NF2 binds phospholipids, promoting its localization to the plasma membrane and activation of Hippo signaling (Hong et al, 2020). NF2 is also reported to act in the nucleus. NF2 inhibits the E3 ligase CRL4[DCAF1], which ubiquitinates and degrades LATS1/2 in the nucleus. Therefore, NF2 stabilizes LATS1/2 in the nucleus, facilitating the phosphorylation and inhibition of YAP at the nuclear level (Li et al, 2014). Another study showed that the formation of a circumferential actin belt at the apical region of cells promotes the interaction between NF2 and YAP in the nucleus, resulting in export of the complex and suppression of YAP activity (Furukawa et al, 2017). It will be interesting to examine whether changes in RAI14 levels have any effect on the previously known role of NF2 in regulating Hippo signaling.

Several studies have shown links between Hippo signaling and GC. Increased nuclear localization of YAP has been identified in GC tissues, and patients with high levels of YAP have a poor prognosis, suggesting that YAP has oncogenic properties

(Kang et al, 2011). Activation of YAP by genetic ablation of *Lats1/2* in *Lgr5*[+] gastric epithelial stem cells initiates gastric tumorigenesis via upregulation of MYC (Choi et al, 2018). Regarding YAP upstream, the levels of Vestigial-like protein 4 (VGLL4), which inhibits YAP-TEAD interaction, are reduced in GC tissue. At the same time, Striatin 3 (STRN3), which promotes dephosphorylation of MST1/2 and activates YAP, is upregulated. In addition, treatment with small peptides that mimic VGLL4 or inhibit STRN3 can block GC progression (Jiao et al, 2014; Tang et al, 2020). In addition, recent studies are shedding light on the relationship between Hippo signaling and DGC. Introduction of a gain-of-function *RHOA* mutation with loss of *CDH1* in *Mist1*[+] gastric stem cells promotes YAP activation and F-actin formation, leading to DGC induction (Zhang et al, 2020). Our clinical data showed that RAI14, YAP, TAZ and F-actin were upregulated in GC tissues, especially in DGC tissues, supporting the previous studies. Therefore, we propose that RAI14 may be one of the players in regulating YAP by responding to tissue stiffness during cancer progression. Further validation, such as the use of RAI14-genetically modified mice and GC model mice, is needed to solidify the role of RAI14 in GC progression.

In conclusion, our study has identified the novel role of RAI14 as a responder to mechanical forces and a regulator of Hippo signaling. This finding suggests that RAI14 may serve as a potential biomarker and therapeutic target for the treatment of GC by modulating Hippo signaling.

# Methods

## Reagents and tools table

| Reagent/resource | Reference or source | Identifier or catalog number |
| --- | --- | --- |
| **Experimental models** | | |
| HEK293T cell line (*H. sapiens*) | American Type Culture Collection (ATCC) | CRL-1573 |
| HEK293A cell line (*H. sapiens*) | Invitrogen | R70507 |
| AGS cell line (*H. sapiens*) | Korean Cell Line Bank (KCLB) | 21739 |
| HCT116 cell line (*H. sapiens*) | Korean Cell Line Bank (KCLB) | 10247 |
| SNU638 cell line (*H. sapiens*) | Korean Cell Line Bank (KCLB) | 00638 |
| MDA-MB-231 cell line (*H. sapiens*) | Korean Cell Line Bank (KCLB) | 30026 |
| C57BL/6NJ-Rai14em1(IMPC)J | Provided by Dr. Sang Ki Park (Pohang University of Science and Technology, Korea) | |
| C57BL/6NJ | Youngbio | |
| **Recombinant DNA** | | |
| pCMV4-Flag | Sigma | |
| pEGFP-C1 | Clontech | |
| Flag-YAP | From Dr. Eek-hoon Jho lab | |
| EGFP-YAP | From Dr. Eek-hoon Jho lab | |
| Flag-RAI14 | This study | |
| EGFP-RAI14 | This study | |
| Flag-TAZ | Provided by Dr. Wantae Kim (University of Seoul, Korea) | |
| Flag-TEAD1 | Provided by Dr. Wantae Kim (University of Seoul, Korea). | |
| Flag-MST1 | Provided by Dr. Dae-Sik Lim (KAIST, Korea) | |
| Flag-MST2 | Provided by Dr. Dae-Sik Lim (KAIST, Korea) | |
| 8XGTIIC-Luc | Addgene | #34615 |
| Myc-LATS1 | Addgene | #41156 |
| Flag-NF2 | Addgene | #107150 |
| HA-Ub | Addgene | #17608 |
| HA-NEDD8 | Addgene | #18711 |
| EGFP-RAI14 (1-424) | This study | |
| EGFP-RAI14 (425-980) | This study | |
| Flag-NF2 (17–595) | This study | |
| Flag-NF2 (1–506) | This study | |
| EGFP-RAI14ΔANK | This study | |
| EGFP-RAI14ΔCC | This study | |
| lentiCRISPR v2 | Addgene | #52961 |
| **Antibodies** | | |
| p-YAP (S127) | Cell signaling | 4911S |
| YAP (for immunoblotting) | Santa cruz | sc-101199 |
| YAP (for immunofluorescence) | Cell signaling | 14074S |
| TAZ | Santa cruz | sc-101199 |
| p-LATS1 (T1079) | Cell signaling | 8654S |
| LATS1 | Cell signaling | 3477S |

| Reagent/resource | Reference or source | Identifier or catalog number |
| --- | --- | --- |
| NF2 (for immunoblotting, proximity-ligation assay) | Cell signaling | 12888S |
| NF2 (for immunofluorescence, proximity-ligation assay) | Santa cruz | sc-55575 |
| RAI14 (for immunoblotting, immunoprecipitation) | Abcam | ab137118 |
| RAI14 (for immunofluorescence, proximity-ligation assay) | Proteintech | 17507-1-AP |
| LC3B | Cell signaling | 2775S |
| p-AKT (S473) | Cell signaling | 9271S |
| EGFP (for immunoblotting) | Santa cruz | sc-9996 |
| EGFP (for immunoprecipitation) | Invitrogen | A-11122 |
| Flag | Sigma | F1804 |
| HA | Santa cruz | sc-7392 |
| myc | Abm | G019 |
| β-actin | Sigma | A5441 |
| Vinculin | Sigma | V9131 |
| IgG rabbit | Cell signaling | 2729S |
| Goat anti-Mouse IgG (HRP conjugate) | Jackson | 115-035-003 |
| Goat anti-Rabbit IgG (HRP conjugate) | Jackson | 111-035-003 |
| Goat anti-Mouse IgG (HRP conjugate, light chain specific) | Jackson | 115-035-174 |
| Mouse anti-Rabbit IgG (HRP conjugate, light chain specific) | Jackson | 211-032-171 |
| Alexa Fluor 488 goat anti-mouse | Invitrogen | A11029 |
| Alexa Fluor 488 goat anti-rabbit | Invitrogen | A11034 |
| Alexa Fluor 546 goat anti-mouse | Invitrogen | A11030 |
| Alexa Fluor 546 goat anti-rabbit | Invitrogen | A11035 |
| **Oligonucleotides and other sequence-based reagents** | | |
| siCtrl | Genolution; this study | Sense: 5′-CCUCGUGCC GUUCCAUCAGGUAGUU-3′ antisense: 5′-CUACCUGAUGGA ACGGCACGAGGUU-3′ |
| siRAI14#1 | Genolution; this study | Sense: 5′-GCUGCUUCUU GCUGUACAAUU-3′ antisense: 5′-UUGUACAGCAA GAAGCAGCUU-3′ |
| siRAI14#2 | Genolution; this study | Sense: 5′-GAUCAGUUCUA UACGAGAAUU-3′ antisense: 5′-UUCUCGUAUAG AACUGAUCUU-3′ |
| siRAI14#3 | Genolution; this study | Sense: 5′-GGAGUUACAAG AUAAAUUAUU-3′ antisense: 5′-UAAUUUAUCUUG UAACUCCUU-3′ |
| siNF2 | Genolution; this study | Sense: 5′-UGGCCAACGA AGCACUGAUUU-3′ antisense: 5′-AUCAGUGCUUC GUUGGCCAUU-3′ |
| shCtrl | Sigma MISSION® shRNA; this study | CCTAAGGTTAA GTCGCCCTCG |
| shRAI14#1 | Sigma MISSION® shRNA; this study | GAAGAGTACGAGGAAATGAAA |

| Reagent/resource | Reference or source | Identifier or catalog number |
|---|---|---|
| shRAI14#2 | This study | CTGATAGCTTATTGGATATAA |
| gRAI14#1 | This study | GAGCTGTCGACACTTTCGGCTGG |
| gRAI14#3 | This study | AGGTCGAACTCCAATCCCGAAGG |
| CTGF | This study | Forward: 5'- AGGAGTGGGT GTGTGACGA-3' reverse: 5'- CCAGGCAGTT GGCTCTAATC-3' |
| CYR61 | This study | Forward: 5'- CCTTGTGGACAG CCAGTGTA-3' reverse: 5'- ACTTGGGCCGG TATTTCTTC-3' |
| ANKRD1 | This study | Forward: 5'- AGTAGAGGAACT GGTCACTGG-3' reverse: 5'- TGGGCTAGAA GTGTCTTCAGAT-3' |
| RAI14 | This study | Forward: 5'- GTGGATGTGA CAGCCCAAGA-3' reverse: 5'- TTTCCCAGAGC TGTCGACAC-3' |
| GAPDH | This study | Forward: 5'- TGCACCACCA ACTGCTTACC-3' reverse: 5'- GGCATGGACT GTGGTCATGAG-3' |
| **Chemicals, enzymes and other reagents** | | |
| Dulbecco's modified eagle medium (DMEM) | Corning | 10-013-CV |
| RPMI-1640 | Hyclone | SH30027 |
| Fetal bovine serum (FBS) | Gibco | 26140-079 |
| Antibiotic-antimycotic | Gibco | 15240-062 |
| Lipofectamine 3000 | Invitrogen | L3000015 |
| Turbofect | Thermo Scientific | R0531 |
| Lipofectamine RNAiMAX | Invitrogen | 13778150 |
| PEG-it Virus Precipitation Solution | SBI System Biosciences | LV810A-1 |
| Polybrene | Sigma | H9268 |
| Puromycin | Sigma | P8833 |
| Protein G agarose beads | Santa Cruz | sc-2003 |
| Pierce™ Protein A/G magnetic beads | Thermo Scientific | #88803 |
| TRI reagent | MRC | TR118 |
| ReverTra AceTM qPCR RT Master Mix | Toyobo | FSQ-201 |
| THUNDERBIRD® SYBR® qPCR Mix | Toyobo | QPS-201 |
| Passive lysis buffer | Promega | E194A |
| Dual-luciferase reporter assay kit | Promega | E1960 |
| 4% paraformaldehyde | Biosesang | P2031 |
| DAPI | Sigma | D9542 |
| Mounting solution | SouthernBiotech | #0100-01 |
| Duolink® In Situ Red Starter Kit Mouse/Rabbit | Sigma | DUO92101 |
| Biotin-XX phalloidin | Thermo Scientific | B7474 |
| HBSS | Gibco | 14175095 |
| Streptavidin —conjugated agarose | Sigma | S1638 |
| Single cell dissociation reagent | STEMCELL Technologies | #100-0485 |

| Reagent/resource | Reference or source | Identifier or catalog number |
|---|---|---|
| DMEM/F12 | Gibco | 11320033 |
| Matrigel | Corning | 356231 |
| IntestiCult™ organoid growth medium | STEMCELL Technologies | #06005 |
| Nicotinamide | Sigma | N0636 |
| Recombinant FGF10 | R&D Systems | 345-FG-025 |
| Recombinant Wnt3A | R&D systems | 5036-WN-010 |
| Gastrin | Sigma | G9020 |
| A8301 | Tocris | 2939 |
| Latrunculin B (Lat. B) | Sigma | L5288 |
| Peterisoft 35 collagen (1 kPa) | Matrigen | PS35-COL-1 |
| Peterisoft 35 collagen (50 kPa) | Matrigen | PS35-COL-50 |
| Y-27632 | SelleckChem | S1049 |
| MG132 | Sigma | C2211 |
| Bafilomycin A1 (Baf. A1) | Santa Cruz | sc-201550 |
| MLN-4924 | SelleckChem | S7109 |
| Alexa Fluor 488 Phalloidin | Invitrogen | A12379 |
| DAPI | Sigma | D9542 |
| Verteporfin | Sigma | SML0534 |
| TRULI | SelleckChem | E1061 |
| MGH-CP1 | SelleckChem | S9735 |
| Lambda Protein Phosphatase | New England Biolabs | P0753S |
| **Software** | | |
| CHOPCHOP | https://chopchop.cbu.uib.no | |
| CellProfiler | https://cellprofiler.org/ | |
| FIJI | https://fiji.sc/ | |
| ColonyArea plug-in | https://imagej.net/plugins/colonyarea | |
| GEPIA2 | http://gepia2.cancer-pku.cn/#index | |
| UCSC Xena | https://xenabrowser.net | |
| OncoLnc | http://www.oncolnc.org | |
| BioRender | https://www.biorender.com/ | |
| GraphPad Prism 10.2.3 | GraphPad | |
| Microsoft Excel | Microsoft | |
| Microsoft PowerPoint | Microsoft | |
| Adobe Illustrator 2024 | Adobe | |
| **Other** | | |
| CytoSoft® 6-well Plates, (0.5 kPa) | Advanced BioMatrix | 5140 |
| CytoSoft® 6-well Plates, (64 kPa) | Advanced BioMatrix | 5145 |
| BD FACSAria™ Fusion Flow Cytometers | BD Biosciences | |
| Microchemi 4.2 | DNR Bio-Imaging Systems | |
| GLOMAX 20/20 | Promega | |
| LSM 800 confocal microscope | Zeiss | |
| CountessTM II cell counter | Invitrogen | |

## Plasmids, siRNAs, gRNAs, shRNAs, and reagents

Human YAP ORF was amplified using cDNA from HEK293T cells and inserted into pCMV4-Flag (Sigma, USA) and pEGFP-C1 (Clontech, USA) to generate Flag-YAP and EGFP-YAP, respectively. To generate Flag-RAI14 and EGFP-RAI14, human RAI14 ORF was amplified with cDNA from HEK293T cells and inserted into pCMV4-Flag (Sigma, USA) and pEGFP-C1 (Clontech, USA), respectively. Flag-TAZ and Flag-TEAD1 were kindly provided by Dr. Wantae Kim (University of Seoul, Korea). Flag-MST1 and Flag-MST2 were kindly provided by Dr. Dae-Sik Lim (KAIST, Korea). 8XGTIIC-Luc, Myc-LATS1, Flag-NF2, HA-Ub, and HA-NEDD8 were obtained from Addgene (plasmid #34615, #41156, #107150, #17608, and #18711 respectively). To generate EGFP-RAI14 (1–424), EGFP-RAI14 (425–980), Flag-NF2 (17–595) and Flag-NF2 (1–506), the truncated regions of RAI14 and NF2 were amplified by PCR and inserted into the appropriate vectors. To generate EGFP-RAI14ΔANK and EGFP-RAI14ΔCC, each deleted region of RAI14 was amplified by PCR splicing method and inserted into pEGFP-C1 (Clontech, USA). To generate the RAI14 guide RNA (gRNA) expressing vector, an annealed gRNA sequence designed using the CHOPCHOP tool (https://chopchop.cbu.uib.no) was inserted into the lentiCRISPR v2 vector (from Addgene, #52961). RAI14 shRNAs were obtained from MISSION® shRNA (Sigma, USA). siRNAs were purchased from Genolution (Korea). Lists of materials and sequences of siRNA, shRNA and gRNA are provided in 'Oligonucleotides and other sequence-based reagents' section of the Reagents and Tools Table.

## Cell culture and transfection

HEK293T cells were obtained from the American Type Culture Collection (ATCC, USA). HEK293A cells were obtained from Invitrogen (USA). AGS, HCT116, SNU638, and MDA-MB-231 cells were obtained from the Korean Cell Line Bank (KCLB, Korea). All cell lines except HEK293A cells were authorized by short tandem repeat analysis, and all cell lines were negative for mycoplasma contamination. All cells were used within 15 passages after thawing. HEK293T and HEK293A cells were cultured in Dulbecco's modified eagle medium (DMEM, Corning) supplemented with 10% fetal bovine serum (FBS, Gibco, 26140-079, USA) and 1X antibiotic-antimycotic (Gibco, 15240-062, USA) at 37 °C in a humidified 5% $CO_2$ incubator. AGS, HCT116, SNU638, and MDA-MB-231 cells were cultured in RPMI-1640 (Gibco) supplemented with 10% fetal bovine serum (FBS, Gibco, 26140-079, USA) and 1X antibiotic-antimycotic (Gibco, 15240-062, USA) at 37 °C in a humidified 5% $CO_2$ incubator. Transfection of plasmid was performed using Lipofectamine 3000 (Invitrogen, L3000015, USA) or Turbofect (Thermo Scientific, R0531, USA) according to the manufacturer's instructions. siRNAs were transfected using Lipofectamine RNAiMAX (Invitrogen, 13778150, USA) according to the manufacturer's instructions.

## Lentiviral transduction and generation of genome-edited cells

HEK293T cells were transfected with shRNA- or gRNA-expressing vector, psPAX2 (packaging vector) and VSV-G (envelope vector) to generate shRNA- or gRNA-containing lentiviruses. One day after transfection, the media was collected, filtered through a 0.45 μm syringe filter and precipitated with PEG-it Virus Precipitation Solution (SBI System Biosciences) according to the manufacturer's instructions. SNU638 cells were treated with shRNA-containing lentivirus in the presence of polybrene (Sigma, 10 μg/ml in media) for one day and selected with puromycin (Sigma, 5 μg/ml in media) for three days. After selection, the cells were used for the appropriate assays. HEK293A cells were treated with gRNA-containing lentivirus in the presence of 10 μg/ml polybrene (Sigma, 10 μg/ml in media) for one day and selected with puromycin (Sigma, 5 μg/ml in media) for three days. After selection, cells were sorted in 96-well plates with a single cell in each well by fluorescence-activated cell sorting (BD FACSAria™ Fusion). RAI14-depleted clones were selected validated by Western blot and used for appropriate assays.

## Lysates preparation, immunoblotting, and immunoprecipitation

Cells were briefly rinsed with ice-cold phosphate-buffered saline (PBS), scraped and centrifuged at 15,000× g, 4 °C for 5 min. Lysis buffer (50 mM Tris-HCl pH 7.4, 150 mM NaCl, 0.5% Triton X-100, 1 mM EDTA, 1 mM EGTA, 2.5 mM sodium pyrophosphate, 1 mM β-glycerophosphate, 1 mM sodium ortho-vanadate, 1 mM phenyl-methylsulfonyl fluoride (PMSF), 1 μg/ml leupeptin) was added to the pellets and incubated on ice for 30 min. The lysates were centrifuged at 15,000× g, 4 °C for 10 min, and the supernatants were collected. Protein concentrations in the supernatants were measured and normalized using the Bradford assay (Bio-Rad, 500-0205, USA). Approximately 20 μg of protein was mixed with 4X dye (200 mM Tris-HCl pH 6.8, 8% SDS, 0.05% bromophenol blue, 40% glycerol, and 200 mM β-mercaptoethanol) and boiled at 95 °C for 10 min.

For immunoblotting, proteins were separated on SDS-polyacrylamide gel electrophoresis (SDS-PAGE) gels (Bio-Rad, #1610154, USA) and transferred to PVDF membrane (ATTO, AE-6667-p, Japan). Membranes were incubated overnight at 4 °C with antibodies diluted in 5% skim milk (Merck, 1.15363.0500, USA) in TBST (10 mM Tris-HCl pH 8.0, 150 mM NaCl, 0.05% TWEEN-20) or 5% bovine serum albumin (BSA, BioBasic, AD0023, Canada) in TBST. After incubation, the membranes were washed five times every 10 min in TBST, and appropriate secondary antibodies diluted in 5% skim milk in TBST were applied to the blots for 1 h at RT. The membranes were additionally washed three times every 10 min in TBST. After washing, signals were detected using enhanced chemiluminescence kits (ELPIS, EBP-1073, Korea, Millipore, WBKLS0100, USA or Thermo Scientific, 34094, USA) and Microchemi 4.2 (DNR Bio-Imaging Systems) according to the manufacturer's instructions. Lists of antibodies are provided in 'Antibodies' section of the Reagents and Tools Table.

For immunoprecipitation, 600–800 μg of lysates were incubated with appropriate antibodies overnight at 4 °C on a rotator. 20 μl Protein G agarose beads (Santa Cruz, sc-2003, USA) or 10 μl Pierce™ Protein A/G magnetic beads (Thermo Scientific, #88803, USA) were added and rotated for 2 h (magnetic beads were used for endogenous immunoprecipitation). To wash the beads, the samples were centrifuged at 3000× g, 4 °C for 2 min and the supernatants were discarded. In total, 800 μl of fresh lysis buffer was added to the beads and the samples were incubated on a rotator at 4 °C for 10 min. After five washes, the beads were mixed with 7 μl of 4X dye and boiled at 95 °C for 10 min. Lists of antibodies are provided in 'Antibodies' section of the Reagents and Tools Table.

## RNA extraction, cDNA synthesis and quantitative real-time PCR

RNA isolation was performed using TRI reagent (MRC, TR118, USA) according to the manufacturer's protocol. RNA was synthesized into cDNA using ReverTra AceTM qPCR RT Master Mix (Toyobo, FSQ-201, Japan) according to the manufacturer's instructions. Quantitative real-time PCR was performed using THUNDERBIRD® SYBR® qPCR Mix (Toyobo, QPS-201, Japan) according to the manufacturer's instructions. The threshold cycle (Ct) value for *GAPDH* was used to normalize the Ct value for each gene, and the ΔΔCt method was used to calculate the relative mRNA expression. The primer sequences used in the experiment are listed in 'Oligonucleotides and other sequence-based reagents' section of the Reagents and Tools Table.

## Luciferase reporter assay

Cells were seeded in a 24-well plate and transfected with mixtures of the desired reporter plasmid (250 ng of 8XGTIIC-Luc in each well of the 24-well plate), thymidine kinase promoter-driven Renilla luciferase (pRL-TK, 25 ng in each well of the 24-well plate) and the desired plasmid constructs or siRNAs. One day later, the cells were treated with 1X passive lysis buffer (PLB, Promega, E194A, USA) for 15 min at RT. The lysates were centrifuged at 7500× *g* for 5 min at 4 °C and 10 μl supernatants were collected for the assay. Firefly and Renilla luciferase activity was analyzed using a dual-luciferase reporter assay kit (Promega, E1960, USA) according to the manufacturer's instructions. GLOMAX 20/20 (Promega, USA) was used to measure luminescence. Firefly luciferase activity was normalized to Renilla luciferase activity.

## Immunofluorescence analysis

Cells on coverslips were fixed with 4% paraformaldehyde (Biosesang, P2031, Korea) for 20 min at RT or 95% methanol (Merck, Germany) for 5 min at RT and washed 3 times with PBS. Cells were permeabilized with 0.1% Triton X-100 in PBS for 15 min at RT and washed 3 times with PBS. After washing, cells were blocked with 5% BSA in PBS for 1 h at RT, and incubated with primary antibodies diluted in 5% BSA in PBS at 4 °C overnight. The antibodies used in the assay are listed in 'Antibodies' section of the Reagents and Tools Table. After incubation, the cells were washed three times with 1% BSA in PBS for 10 min, and incubated with appropriate secondary antibodies diluted in 5% BSA in PBS for 1 h at RT. After incubation, the cells were washed 3 times with 1% BSA in PBS for 10 min. DAPI (300 ng/ml) (Sigma, D9542) staining was performed during the second wash. Glass coverslips were mounted with mounting solution (SouthernBiotech, #0100-01, USA) and fluorescence was measured using a LSM 800 confocal microscope (Zeiss). Quantification of the nuclear/cytoplasmic ratio of YAP in individual cells was performed using the open-source software CellProfiler.

## Proximity-ligation assay

Cells on coverslips were fixed with 4% paraformaldehyde (Biosesang, P2031, Korea) for 20 min at RT or 95% methanol (Merck, Germany) for 5 min at RT and washed three times briefly with PBS. Cells were permeabilized with 0.1% Triton X-100 in PBS for 15 min at RT and washed three times with PBS. After washing,

the cells were blocked with Duolink® blocking solution for 1 h at 37 °C in a heated humidity chamber. After removal of the blocking solution, the cells were incubated with primary antibodies diluted in Duolink® antibody diluent overnight at 4 °C on a rocker. After incubation, the cells were washed twice for 5 min with 1X Duolink® in situ wash buffer A at RT and incubated with Duolink® PLA probes for 1 h at 37 °C in a heated humidity chamber. After incubation, the cells were washed twice for 5 min with 1X Duolink® in situ wash buffer A at RT and incubated with Duolink® ligation buffer for 30 min at 37 °C in a heated humidity chamber. The cells were then washed twice for 5 min with 1X Duolink® In situ wash buffer A at RT and incubated with amplification buffer for 100 min at 37 °C in a heated humidity chamber. The cells were washed twice for 10 min with 1X Duolink® In situ wash buffer B at RT and the final wash step was performed with 0.01X Duolink® In situ wash buffer B at RT. After washing, the remaining buffer was removed and the cells were mounted using Duolink® In situ mounting medium with DAPI. PLA signals were measured using an LSM 800 confocal microscope (Zeiss). Quantification of PLA signals in each individual cell was performed in FIJI using the 'threshold' and 'analyze particle' plug-ins. Lists of antibodies are provided in 'Oligonucleotides and other sequence-based reagents' section of the Reagents and Tools Table.

## F-actin binding assay

We followed the protocol published by Stefano Piccolo's group (Chang et al, 2018). Cells were treated with Lat. B (0.5 μM) or biotin-XX phalloidin (40 ng/ml, Thermo Scientific, B7474, USA) for 4 h. After treatment, cells were rinsed with HBSS (Gibco, USA) and lysed in actin lysis buffer (20 mM HEPES (pH 7.5), 50 mM KCl, 0.1% Triton X-100, 5% glycerol, 0.1% NP-40, 5 mM MgCl$_2$, 1 μM DTT, 10 μM MG132, 1 mM ATP, 20 μM phosphocreatine di(Tris) salt (P1937, Sigma-Aldrich), 2.5 mM sodium pyrophosphate, 1 mM β-glycerophosphate, 1 mM phenylmethylsulfonyl fluoride (PMSF) and 1 μg/ml leupeptin). All buffers were freshly prepared and pre-warmed at RT. Lat. B-treated cells were lysed in actin lysis buffer containing 1 μM Lat. B. Biotin-XX phalloidin-treated cells were lysed in actin lysis buffer containing 40 ng/ml biotin-XX phalloidin. Lysates were scraped, passed through a 26-gauge needle ten times and incubated at RT for 30 min. After incubation, the lysates were centrifuged at 10,000× *g* for 10 min at RT. The supernatants were collected and 30 μl streptavidin-conjugated agarose (Sigma, S1638, USA) was added and rotated at RT for 3 h. During incubation, 1 μg/ml biotin-XX phalloidin was maintained in the supernatants. After incubation, the samples were centrifuged at 3000× *g*, RT for 2 min, and the supernatants were discarded. In all, 800 μl fresh actin lysis buffer was added to the beads and the samples were incubated on a rotator at RT for 10 min. After 3 washes, the beads were mixed with 7 μl of 4X dye and boiled at 95 °C for 10 min.

## Colony-forming assay

For HEK293A cells, transfected cells were trypsinized, and the number of cells was counted using a CountessTM II cell counter (Invitrogen, USA). Briefly, 500 cells were seeded in each well of a 12-well plate and incubated in DMEM supplemented with 10% FBS for approximately 10 days at 37 °C in a humidified 5% CO$_2$

incubator. After incubation, the medium was removed, and staining solution (95% MeOH, 0.5% acetic acid, 0.5% crystal violet) was added to each well for 20 min. The staining solution was removed and the wells were gently washed with tap water. Images for triplicate wells were captured using a digital camera. Colony-covered surface areas were analyzed using FIJI with the 'ColonyArea' plug-in. For SNU638 cells, 5,000 cells were seeded in each well of a 12-well plate. Cells were incubated for approximately 7 days at 37 °C in a humidified 5% $CO_2$ incubator. The remaining steps were the same as for HEK293A cells.

## Cell proliferation assay

For HEK293A and SNU638 cells, transfected cells were trypsinized, and the number of cells was counted using a CountessTM II cell counter (Invitrogen, USA). 5,000 cells were seeded in each well of a 24-well plate (for days 2 and 4) and a 12-well plate (for days 6). HEK293A and SNU638 cells were incubated in DMEM and RPMI-1640 supplemented with 10% FBS at 37 °C in a humidified 5% $CO_2$ incubator for the indicated days. The number of cells in triplicate wells was counted every 2 days for 6 days using a CountessTM II cell counter (Invitrogen, USA).

## TCGA database analysis

The gene expression profiling interactive analysis 2 (GEPIA2, http://gepia2.cancer-pku.cn/#index) website was used to search for differentially expressed genes that have a poor prognosis when expressed at high levels in GC patients. The UCSC Xena website (https://xenabrowser.net) was used to investigate correlations between the expression of *RAI14* and YAP target genes (*CTGF*, *CYR61*). The UCSC Xena website (https://xenabrowser.net) was used to retrieve RAI14 expression data from normal and tumor tissue, and the OncoLnc website (http://www.oncolnc.org) was used to generate survival curve data.

## ACRG database analysis

To evaluate the prognostic impact of *RAI14* in GC patients, we analyzed genome-wide expression profiling data from publicly available datasets (GSE62254 and GSE66222). The data were downloaded from the Gene Expression Omnibus (GEO) dataset. The GSE62254 cohort included 300 GC cases and GSE66222 included expression data from 100 matched normal tissues.

## Preparation of mice gastric tissues

Live perinatal mice were sacrificed by inhalation of carbon dioxide and gastric tissues were dissected. Tissues were homogenized in lysis buffer (50 mM Tris-HCl pH 7.4, 150 mM NaCl, 1% Triton X-100, 1 mM EDTA, 1 mM EGTA, 2.5 mM sodium pyrophosphate, 1 mM β-glycerophosphate, 1 mM sodium ortho-vanadate, 1 mM phenylmethylsulfonyl fluoride (PMSF), 1 μg/ml leupeptin) at 4 °C and centrifuged at 12,000× *g*, 4 °C for 15 min. The supernatant was collected and centrifuged again at 12,000× *g*, 4 °C for 10 min. The supernatant was collected and analyzed for immunoblotting. All experimental procedures were performed in accordance with animal care and ethical guidelines. The protocols were reviewed

and approved by the Institutional Animal Care and Use Committee (IACUC) of the University of Seoul (UOS IACUC-2023-02).

## Generation of mouse intestinal and gastric organoids

To generate mouse intestinal organoids, the mouse was sacrificed by carbon dioxide inhalation and the upper part of the small intestine was collected. The tissues were washed and rinsed with PBS using a pipette and cut longitudinally. The villus part of the small intestine was gently scraped three times and removed. The tissues were cut several times at 2 mm and placed in a 50 ml conical tube containing 15 ml DPBS. The tissues were pipetted three times with pre-wetted serological pipettes and allowed to fall by gravity. The supernatants were removed and replaced with 15 ml DPBS and pipetted 15–20 times or until the supernatants were clean. After letting down, the supernatants were removed and supplemented with single cell dissociation reagent (STEMCELL technologies, #100-0485, USA) and incubated on a rocker for 15 min. After letting down, the supernatants were removed and supplemented with 0.1% BSA in DPBS, pipetted three times and centrifuged at 200× *g* for 3 min at 4 °C. The pellets were pipetted with DMEM/F12 (Gibco) and approximately 1000 crypts were transferred to a conical tube and centrifuged at 200× *g* for 5 min at 4 °C. The supernatants were removed and the crypts were mixed with Matrigel (Corning, USA), transferred to a 24-well plate and incubated at 37 °C for 10 min. Pre-warmed IntestiCult™ organoid growth medium (STEMCELL technologies, #06005, USA) was added in dome-shaped solidified gels. The medium was changed three times a week and the organoids were passaged weekly.

To generate gastric organoids, the corpus part of the stomach was collected. Mucus and muscle parts were carefully removed and washed with PBS. The tissues were cut several times at 5 mm and placed in a 50 ml conical tube containing 15 ml DPBS. The tissues were pipetted three times with pre-wetted serological pipettes and allowed to fall by gravity. The supernatants were removed and replaced with 15 ml DPBS and pipetted 15–20 times or until the supernatants were clean. After letting down, the supernatants were removed and supplemented with single cell dissociation reagent (STEMCELL technologies, #100-0485, USA) and incubated on a rocker for 15 min. The tissue was collected, placed on a 100 mm plate and pressed through microscope glass. The pressed tissue was collected in a conical tube containing 10 ml DMEM/F12, pipetted three times, and approximately 100 glands were transferred to a conical tube and centrifuged at 200 × *g* for 5 min at 4 °C. Supernatants were removed and crypts were mixed with Matrigel (Corning, USA), transferred to a 24-well plate and incubated at 37 °C for 10 min. Pre-warmed IntestiCult™ Organoid Growth Medium (STEMCELL technologies, #06005, USA) supplemented with 1 mM nicotinamide (Sigma, USA), 200 ng/ml recombinant FGF10 (R&D Systems, USA), 60 ng/ml recombinant Wnt3A (R&D systems, USA), 200 ng/ml gastrin (Sigma, USA), 2 μM A8301 (Tocris, UK) and 10 μM Y-27632 (Selleckchem, USA) in dome-shaped solidified gels. Medium was changed three times a week and organoids were passaged weekly.

All experimental procedures were performed in accordance with animal care and ethics guidelines; the protocols were reviewed and approved by the Institutional Animal Care and Use Committee (IACUC) of the University of Seoul (UOS IACUC-2023-03).

## Tissues from human gastric patients

For clinical validation, a total of 16 patients with GC (8 diffuse type and 8 intestinal type tumors based on Lauren classification) were enrolled at the University of Ulsan and Asan Medical Center, Seoul, Korea. They underwent curative surgery for biopsy-proven primary GC between 2011 and 2018. Cancerous tissues were obtained during surgery from a representative malignant lesion in the surgically resected specimen, and non-cancerous tissues were obtained from the area farthest away from the cancer. They were immediately snap-frozen in liquid nitrogen and stored at −80 °C at the Asan Bio-Resource Center. All procedures were performed in accordance with the Helsinki Declaration. Written informed consent was obtained from all participants. This study was approved by the Institutional Review Boards of Asan Medical Center, Seoul, Korea (IRB No. 2023-0459).

## Graphics

Figures 5F and 6A, Fig. 8 and the synopsis image were generated using BioRender.com.

## Statistics and data processing

Statistical analyses were performed using two-tailed unpaired *t*-test and log-rank test, or calculating the right-tailed F probability distribution with Microsoft Excel or GraphPad Prism software. A *P* value of less than 0.05 was considered statistically significant. Quantitative data from three replicates are presented as mean ± s.d. Quantitative data from the cell image are presented as mean ± s.e.m. The hazard ratio (HR) of the survival plots was calculated using the Mantel–Haenszel method. The correlation coefficient (*r*) of scatter plots was calculated using Pearman's linear correlation. Graphical data were processed and visualized using GraphPad Prism software, and all figures were processed using Microsoft PowerPoint and Adobe Illustrator software.

# Data availability

No primary datasets have been generated and deposited.

The source data of this paper are collected in the following database record: biostudies:S-SCDT-10_1038-S44319-024-00228-0.

# Peer review information

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

## Acknowledgements

This work was supported by an NRF grant funded by the MIST to E Jho (RS-2023-00208583). NRF grants funded by the MOE supported W Jeong (2021R1A6A3A13045151 and RS-2023-00275628) and H Kwon

(2021R1A6A3A13046086). The Brain Korea 21 program supported W Jeong and H Kwon. Asan Bio-Resource Center, Korea Biobank Network [2023-16(267)] provided the biospecimen and data used in this study. I Lee was supported by sponsored research funds provided by Mr. Hyeonho Kim.

## Author contributions

**Wonyoung Jeong**: Conceptualization; Funding acquisition; Investigation; Writing—original draft; Writing—review and editing. **Hyeryun Kwon**: Conceptualization; Investigation; Writing—original draft; Writing—review and editing. **Sang Ki Park**: Resources; Investigation. **In-Seob Lee**: Conceptualization; Resources; Investigation; Writing—original draft; Writing—review and editing. **Eek-hoon Jho**: Conceptualization; Supervision; Funding acquisition; Writing—original draft; Writing—review and editing.

Source data underlying figure panels in this paper may have individual authorship assigned. Where available, figure panel/source data authorship is listed in the following database record: biostudies:S-SCDT-10_1038-S44319-024-00228-0.

## Disclosure and competing interests statement

The authors declare no competing interests.

# Expanded View Figures

**Figure EV1. Finding novel regulators of Hippo signaling by analyzing the TCGA database.**

(A) List of the top fifteen differentially expressed genes with poor prognosis at high expression levels in stomach adenocarcinoma (STAD). Genes with a high correlation with YAP target genes (*CTGF* and *CYR61*) are highlighted in light yellow. Genes with increased expression in tumors are highlighted in apricot. Genes corresponding to both of the above conditions are highlighted in yellow. (B) Kaplan–Meier plots show that high expression of *RAI14* in gastric cancer patients is associated with poor prognosis. The upper 50th percentile ($n = 189$) and the lower 50th percentile ($n = 189$) were analyzed in gastric cancer patients. HR, hazard ratio. (C) The violin plot shows a high expression of RAI14 in gastric tumor samples. The Y-axis indicates the log base 2 of RAI14 expression. The analysis included normal samples from the GTEx database ($n = 172$) and the TCGA database ($n = 36$) as well as tumor samples from the TCGA database ($n = 414$). (D) The mRNA expression of gastric cancer patients ($n = 450$) from the TCGA database shows a positive correlation between *RAI14* and YAP-target genes (*CTGF* and *CYR61*). Data information: In (B), the hazard ratio (HR) was calculated using the Mantel–Haenszel method. In (D), the correlation coefficient (r) was calculated using Pearman's linear correlation. Statistical analysis was performed using log-rank test (B) and two-tailed unpaired *t* test (C), or calculating right-tailed F probability distribution (D), and exact *P* values are shown in each figure; $P < 0.05$, statistically significant. Source data are available online for this figure.

▶

**A**

Top15 differential Survival Genes in STAD (gastric cancer)

| RANK | Gene name | Gene ID | P-Value (Survival os) | Prognosis | Correlation with *CTGF* (Pearson correlation R) | Correlation with *CYR61* (Pearson correlation R) | Increased expression in tumor? |
|---|---|---|---|---|---|---|---|
| 1 | *GFAP* | ENSG00000131095.11 | $2.12 \times 10^{-5}$ | unfavorable | 0.23 | 0.2 | |
| 2 | *RP11-497E19.1* | ENSG00000205562.2 | $2.33 \times 10^{-5}$ | unfavorable | 0.28 | 0.3 | |
| 3 | *ASPA* | ENSG00000108381.10 | $2.69 \times 10^{-5}$ | unfavorable | 0.42 | 0.36 | |
| 4 | *SERPINE1* | ENSG00000106366.8 | $3.41 \times 10^{-5}$ | unfavorable | 0.39 | 0.43 | O |
| 5 | *ZNF883* | ENSG00000228623.3 | $3.59 \times 10^{-5}$ | unfavorable | -0.026 | -0.0087 | |
| 6 | *AOC4P* | ENSG00000260105.6 | $3.86 \times 10^{-5}$ | unfavorable | 0.36 | 0.31 | |
| 7 | *CBLN4* | ENSG00000054803.3 | $3.97 \times 10^{-5}$ | unfavorable | 0.11 | 0.15 | |
| 8 | *NT5E* | ENSG00000135318.11 | $4.74 \times 10^{-5}$ | unfavorable | 0.082 | 0.044 | O |
| 9 | *AC002480.3* | ENSG00000232759.1 | $6.37 \times 10^{-5}$ | unfavorable | 0.062 | 0.11 | |
| 10 | *MEI4* | ENSG00000269964.2 | $6.39 \times 10^{-5}$ | unfavorable | 0.065 | 0.054 | |
| 11 | *CTD-2054N24.2* | ENSG00000259363.5 | $6.93 \times 10^{-5}$ | unfavorable | 0.3 | 0.24 | |
| 12 | *RP11-1069G10.2* | ENSG00000259727.1 | $9.09 \times 10^{-5}$ | unfavorable | 0.23 | 0.18 | |
| 13 | *EMX2OS* | ENSG00000229847.8 | $9.24 \times 10^{-5}$ | unfavorable | 0.16 | 0.16 | |
| 14 | *ZNF192P1* | ENSG00000226314.7 | $9.45 \times 10^{-5}$ | unfavorable | 0.023 | -0.0096 | |
| 15 | *RAI14* | ENSG00000039560.13 | $9.67 \times 10^{-5}$ | unfavorable | 0.49 | 0.4 | O |

☐ Genes having a high correlation with *CTGF*, *CYR61* (R > 0.3)

☐ Genes with increased expression in tumors

☐ Genes having a high correlation with *CTGF*, *CYR61* and increased expression in tumors

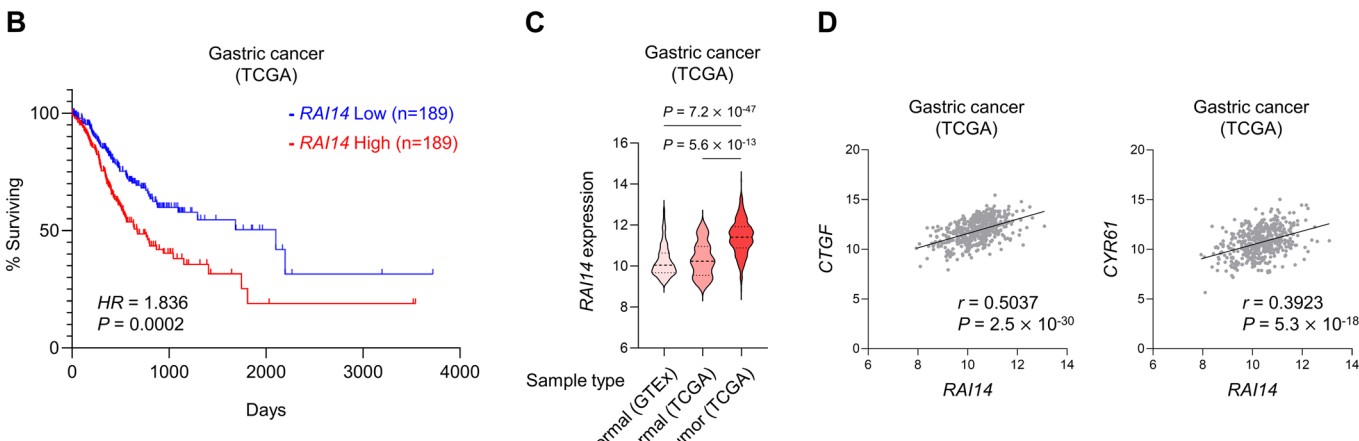

**B** Gastric cancer (TCGA)

- *RAI14* Low (n=189)
- *RAI14* High (n=189)

HR = 1.836
P = 0.0002

**C** Gastric cancer (TCGA)

P = $7.2 \times 10^{-47}$
P = $5.6 \times 10^{-13}$

**D** Gastric cancer (TCGA)

r = 0.5037
P = $2.5 \times 10^{-30}$

Gastric cancer (TCGA)

r = 0.3923
P = $5.3 \times 10^{-18}$

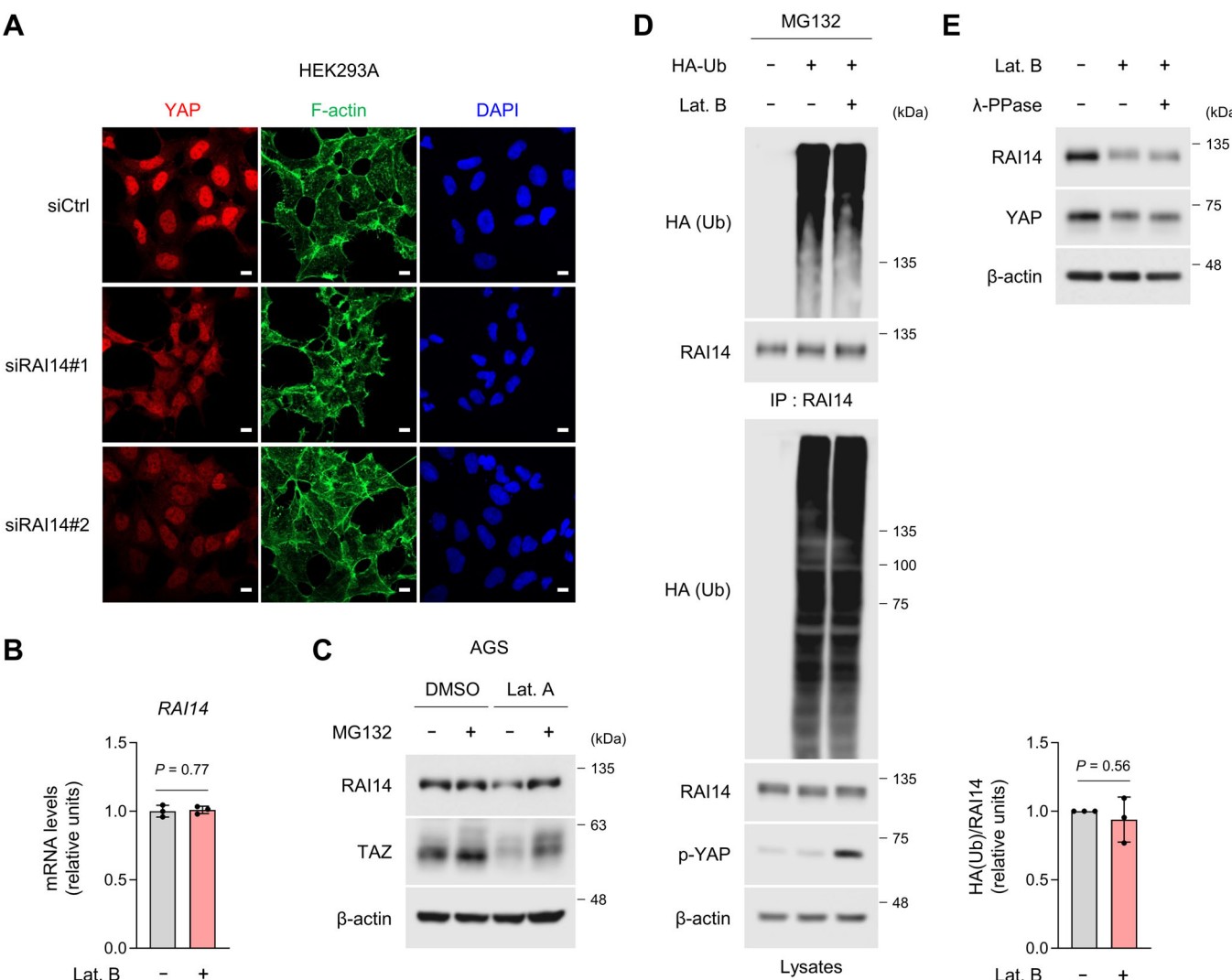

**Figure EV2. Destabilization of F-actin induces proteasomal degradation of RAI14.**

(A) Marginal effect of RAI14 knockdown on F-actin formation. HEK293A cells were used. For F-actin staining, phalloidin was used. Scale bars: 10 μm. (B) Destabilization of F-actin by Latrunculin B (Lat. B) does not reduce RAI14 mRNA expression. HEK293A cells were treated with DMSO or Lat. B (2 μM) for 6 h. (C) F-actin destabilization-mediated downregulation of RAI14 is dependent on the proteasomal degradation pathway. AGS cells were treated with Lat. A (2 μM) and MG132 (25 μM) for 6 h as indicated in the figure. (D) Destabilization of F-actin does not induce polyubiquitination of RAI14. HEK293A cells were transfected and treated with Lat. B (2 μM) and MG132 (25 μM) for 6 h as indicated in the figure. The HA(Ub)/RAI14 ratio from three immunoblot bands was quantified. (E) The mobility shift of RAI14 by F-actin destabilization is due to its phosphorylation. HEK293A cells were treated with Lat. B for 6 h and lambda phosphatase (λ-PPase) according to the manufacturer's instructions. Data information: In (B, D), the error bars indicate ± s.d. of triplicate measurements (B is technical replicates, and D is biological replicates). Statistical analysis was performed using two-tailed unpaired *t* test, and exact *P* values are shown in each figure; *P* < 0.05, statistically significant. Black dots on the graphs indicate individual measurements. Source data are available online for this figure.

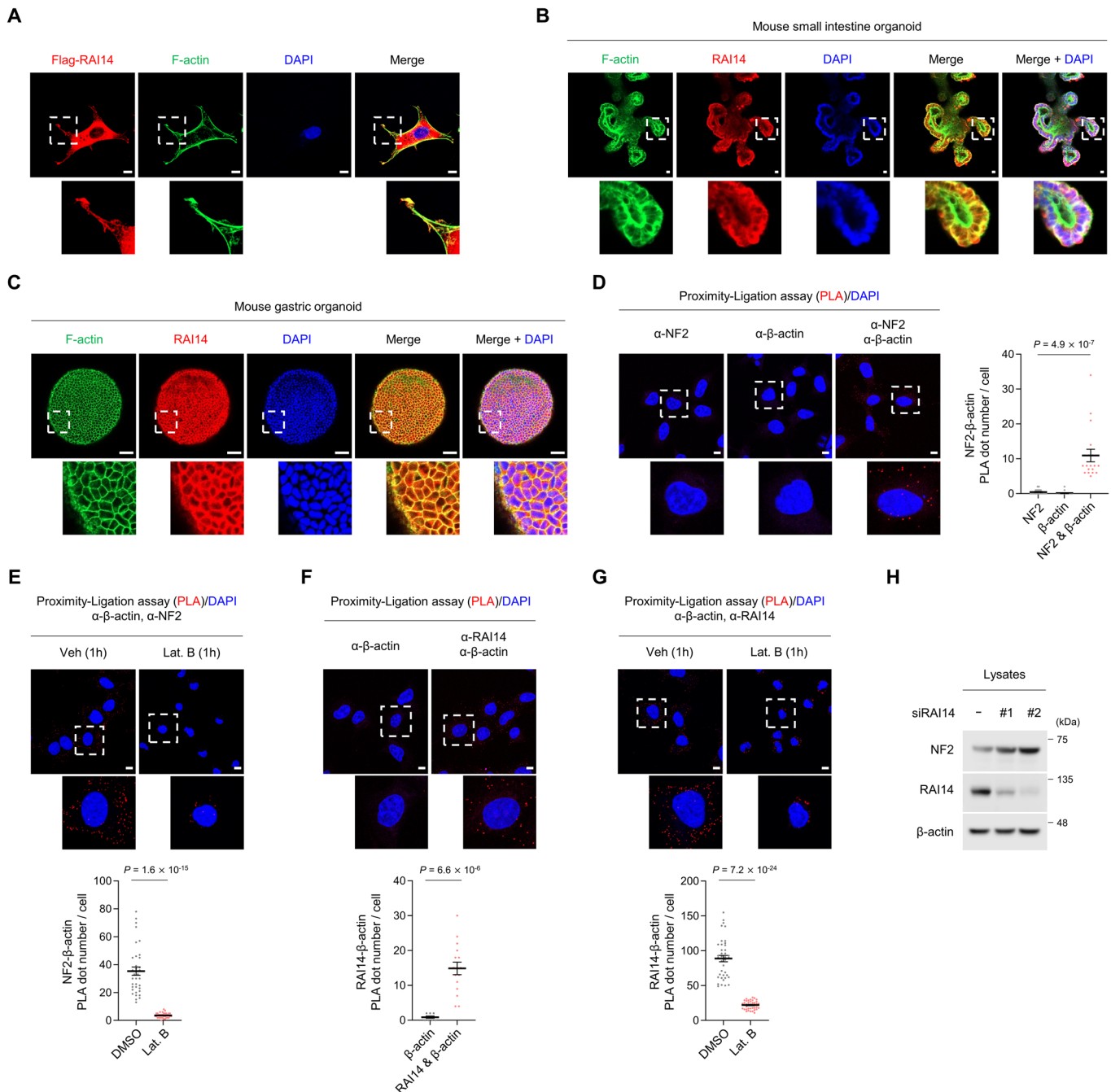

**Figure EV3. RAI14 and NF2 interaction occurs on F-actin.**

(A) Overexpressed Flag-RAI14 colocalizes with F-actin. HEK293A cells were used. Scale bars: 10 μm. (B, C) RAI14 colocalizes with F-actin at the endogenous level in organoid models. Mouse small intestine organoids (B, scale bars: 10 μm) or gastric organoids (C, scale bars: 50 μm) were used. (D–G) Endogenous NF2 and β-actin (D, E), and RAI14 and β-actin (F, G) are in proximity. HEK293A cells were treated with Lat. B (2 μM) for 1 h. The quantification of the number of dots in a single cell is shown. In (D), 20, 21, and 18 cells were used for NF2, β-actin, and NF2&β-actin, respectively. In (E), 35 and 33 cells were used for DMSO and Lat. B, respectively. In (F), 9 and 16 cells were used for β-actin and RAI14&β-actin, respectively. In (G), 40 cells were used for DMSO and Lat. B, respectively. Scale bars: 10 μm. (H) Immunoblot of lysates used for Fig. 5E. Data information: In (D–G), the error bars indicate ± s.e.m. of the measurements (technical replicates). Statistical analysis was performed using two-tailed unpaired t-test, and exact P values are shown in each figure; P < 0.05, statistically significant. Colored dots on the graphs indicate individual measurements. Source data are available online for this figure.

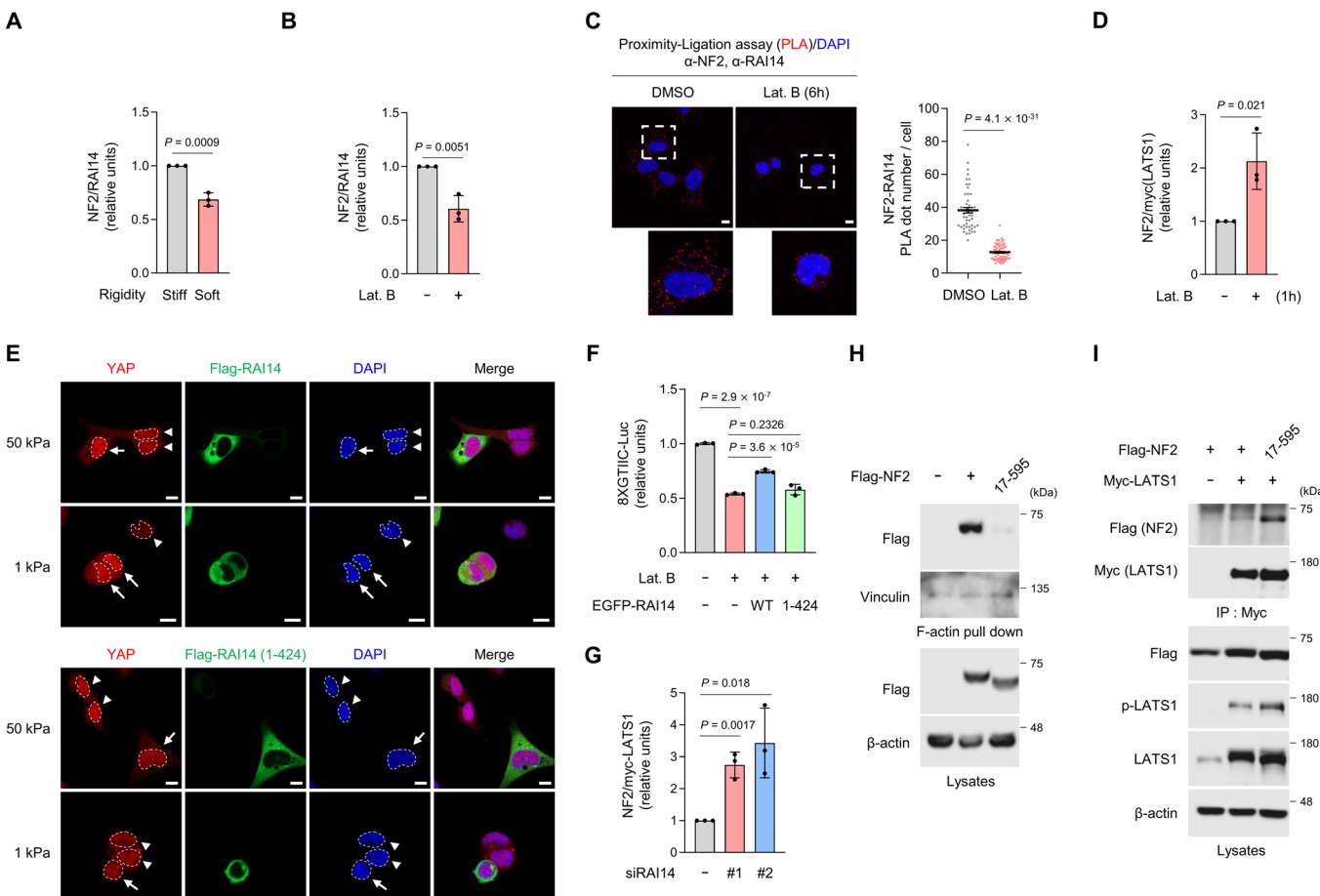

**Figure EV4. ECM stiffness and F-actin integrity regulate the interaction between RAI14, NF2 and LATS.**

(A, B) Quantification of immunoprecipitated NF2/RAI14 ratio from three immunoblot bands in Fig. 6B,C, respectively. (C) PLA assays show that the interaction between NF2 and RAI14 is inhibited by Lat. B treatment. HEK293A cells were treated with Lat. B (2 μM) and MG132 (25 μM) for 6 h. The quantification of the number of dots in a single cell is shown in the right panel. 52 and 70 cells were used for DMSO and Lat. B, respectively. Scale bars: 10 μm. (D) Quantification of immunoprecipitated NF2/Myc(LATS1) ratio from three immunoblot bands in Fig. 6H. (E) Overexpression of RAI14, but not the RAI14(1–424) mutant form, promotes nuclear localization of YAP under both hard and soft matrix conditions. HEK293A cells were transfected and incubated on a stiff (50 kPa) or soft (1 kPa) matrix for one day. Nuclei of Flag-RAI14 (upper panel) or Flag-RAI14(1–424) (lower panel) overexpressing cells are marked with a white arrow. Nuclei of un-transfected cells (upper and lower panel) are marked with a white arrowhead. Scale bars: 10 μm. (F) Overexpression of RAI14 partially rescues the YAP reporter activity but not in RAI14 (1–424) form. HEK293A cells were transfected and treated with Lat. B (2 μM) for 6 h. (G) Quantification of immunoprecipitated NF2/Myc(LATS1) ratio from three immunoblot bands in Fig. 6K. (H) The NF2 (17–595) form fails to bind F-actin. HEK293A cells were used. (I) The NF2 (17–595) form shows increased interaction with LATS1 and phosphorylation of LATS1 compared to wild-type NF2. HEK293A cells were used. Data information: In (A, B, D, F, G), the error bars indicate ± s.d. of triplicate measurements (biological replicates). In (C), the error bars indicate ± s.e.m. of the measurements (technical replicates). Statistical analysis was performed using two-tailed unpaired *t* test, and exact *P* values are shown in each figure; *P* < 0.05, statistically significant. Black or colored dots on the graphs indicate individual measurements. Source data are available online for this figure.

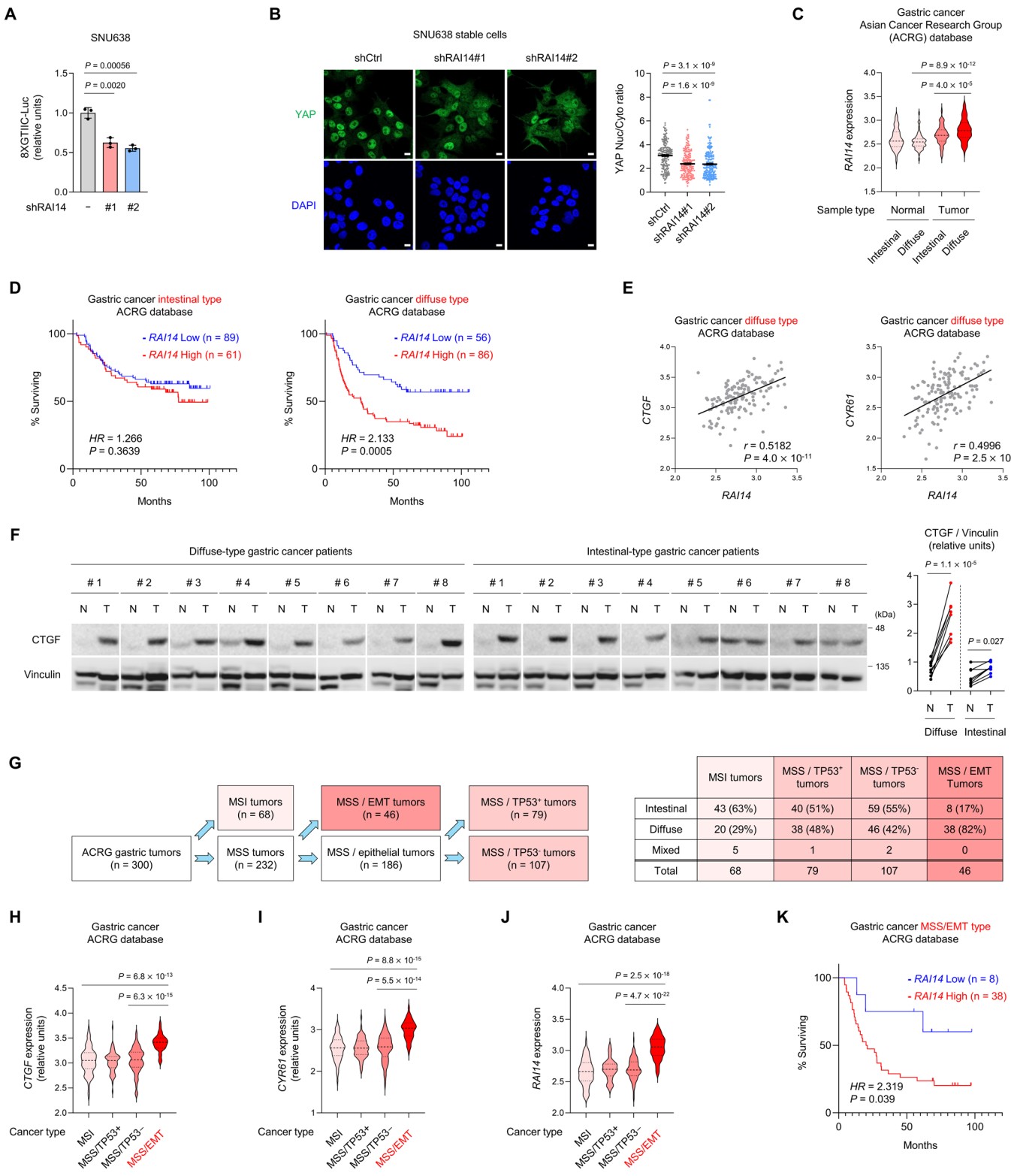

**Figure EV5.   RAI14-Hippo signaling promotes cell proliferation and growth, and is implicated in gastric cancer progression.**

(A) Downregulation of RAI14 by shRNA reduces YAP-reporter activity. SNU638 cells were used. (B) Knockdown of RAI14 reduces the nuclear localization of YAP. SNU638 cells were used. Scale bars: 10 μm. Quantification of the nuclear/cytoplasmic ratio in a single cell is shown in the right panel. 170, 187, and 203 cells were used for shCtrl, shRAI14#1, and shRAI14#2, respectively. (C) The violin plot shows a high expression of *RAI14* in gastric tumor samples, especially in the diffuse-type. The y-axis indicates the log base 10 of *RAI14* expression. Adjacent normal tissue samples from intestinal-type ($n = 39$) and diffuse-type ($n = 58$) patients and tumor tissue samples from intestinal-type ($n = 150$) and diffuse-type ($n = 142$) patients were analyzed. (D) Kaplan–Meier plots show that high expression of *RAI14* in diffuse-type GC patients is associated with poor prognosis. The upper 41st ($n = 61$) and lower 59th ($n = 89$) percentiles were analyzed for intestinal-type GC patients, and the upper 61st ($n = 86$) and lower 39th ($n = 56$) percentiles were analyzed for diffuse-type GC patients. HR, hazard ratio. (E) The mRNA expression of diffuse-type GC ($n = 142$) from the ACRG database shows a positive correlation between *RAI14* and YAP-target genes (*CTGF* and *CYR61*). (F) CTGF level were high in GC tissues and significantly higher in DGC. Quantification of CTGF/Vinculin ratios from immunoblots was performed as in Fig. 7K. (G) Schematic representation of the molecular classification of GC from the ACRG database (left panel). Patients with MSS/EMT-type mainly belong to the diffuse-type (right panel). (H–J) The violin plot shows high expression of *CTGF, CYR61* and *RAI14* in MSS/EMT-type gastric tumor samples. The y-axis indicates the log base 10 of *RAI14* expression. Tissue samples from MSI ($n = 68$), MSS/TP53$^+$ ($n = 79$), MSS/TP53$^-$ ($n = 107$) and MSS/EMT-type patients ($n = 46$) were analyzed. (K) Kaplan–Meier plots show that high expression of *RAI14* in MSS/EMT-type GC patients is associated with a worse prognosis. The upper 17th percentile ($n = 8$) and the lower 83rd percentile ($n = 38$) were analyzed in patients with intestinal-type gastric cancer. HR, hazard ratio. Data information: In (A), the error bars indicate ± s.d. of triplicate measurements (biological replicates). In (B), the error bars indicate ± s.e.m. of the measurements (technical replicates). In (D, K), the hazard ratio (HR) was calculated using the Mantel–Haenszel method. In (E), the correlation coefficient (r) was calculated using Pearman's linear correlation. Statistical analysis was performed using two-tailed unpaired *t* test (A–C, F, H–J) and log-rank test (D, K), or calculating right-tailed F probability distribution (E), and exact *P* values are shown in each figure; $P < 0.05$, statistically significant. Black or colored dots on the graphs indicate individual measurements. Source data are available online for this figure.

