## [Peer Review File · EMBO Reports]

Retinoic acid-induced protein 14 links mechanical forces to Hippo signaling

Wonyoung Jeong, Hyeryun Kwon, Sang Ki Park, In-Seob Lee, and Eek-hoon Jho

Corresponding authors: Eek-hoon Jho (ej70@uos.ac.kr) , In-Seob Lee (inseoble77@amc.seoul.kr)

Review Timeline:

Submission Date:	18th Mar 24
Editorial Decision:	8th Apr 24
Revision Received:	8th Jul 24
Editorial Decision:	22nd Jul 24
Revision Received:	29th Jul 24
Accepted:	5th Aug 24

Editor: Achim Breiling

Transaction Report:

Dear Prof. Jho,

Thank you for the transfer of your research manuscript to EMBO reports. I have now received the reports from the three referees that were asked to evaluate your study, which can be found at the end of this email.

As you will see, the referees think that the findings are of high interest. Nevertheless, they have several comments, concerns, and suggestions, indicating that a major revision of the manuscript is necessary to allow publication of the study in EMBO reports. As the reports are below, and all the concerns need to be addressed, I will not detail them here.

Given the constructive referee comments, I would like to invite you to revise your manuscript with the understanding that the concerns of the referees must be addressed in the revised manuscript and in a detailed point-by-point response. Acceptance of your manuscript will depend on a positive outcome of a second round of review. It is EMBO reports policy to allow a single round of revision only and acceptance of the manuscript will therefore depend on the completeness of your responses included in the next, final version of the manuscript.

- 1) a .docx formatted version of the final manuscript text (including legends for main figures, EV figures and tables), but without the figures included. Figure legends should be compiled at the end of the manuscript text.
- 2) individual production quality figure files as .eps, .tif, .jpg (one file per figure), of main figures and EV figures. Please upload these as separate, individual files upon re-submission.

- 4) a complete author checklist, which you can download from our author guidelines (<https://www.embopress.org/page/journal/14693178/authorguide>). Please insert page numbers in the checklist to indicate where the requested information can be found in the manuscript. The completed author checklist will also be part of the RPF.

- 5) that primary datasets produced in this study (e.g. RNA-seq, ChIP-seq, structural and array data) are deposited in an

appropriate public database. If no primary datasets have been deposited, please also state this in a dedicated section (e.g. 'No primary datasets have been generated and deposited'), see below.

The accession numbers and database should be listed in a formal "Data Availability" section (placed after Materials & Methods) that follows the model below. This is now mandatory (like the COI statement). Please note that the Data Availability Section is restricted to new primary data that are part of this study. This section is mandatory. As indicated above, if no primary datasets have been deposited, please state this in this section

Data availability

8) Regarding data quantification and statistics, please make sure that the number "n" for how many independent experiments were performed, their nature (biological versus technical replicates), the bars and error bars (e.g. SEM, SD) and the test used to calculate p-values is indicated in the respective figure legends (also for EV figures and all those in an Appendix). Please also check that all the p-values are explained in the legend, and that these fit to those shown in the figure. Please provide statistical testing where applicable. Please avoid the phrase 'independent experiment', but clearly state if these were biological or technical replicates. Please also indicate (e.g. with n.s.) if testing was performed, but the differences are not significant. In case n=2, please show the data as separate datapoints without error bars and statistics. See also: <http://www.embopress.org/page/journal/14693178/authorguide#statisticalanalysis>

Moreover, please add a 'Data Information' section to each figure legend (main and EV figures) explaining the statistics used or providing information regarding replicates and scales. See:

9) Please add scale bars of similar style and thickness to microscopic images, using clearly visible black or white bars (depending on the background). Please place these in the lower right corner of the images themselves. Please do not write on or near the bars in the image but define the size in the respective figure legend.

10) Please also note our reference format:

12) We now use CRediT to specify the contributions of each author in the journal submission system. CRediT replaces the

author contribution section. Please use the free text box to provide more detailed descriptions and do not provide your final manuscript text file with an author contributions section. See also our guide to authors:
<https://www.embopress.org/page/journal/14693178/authorguide#authorshipguidelines>

13) We would encourage you to use 'Structured Methods', our new Materials and Methods format. According to this format, the Materials and Methods section should include a Reagents and Tools Table (listing key reagents, experimental models, software, and relevant equipment and including their sources and relevant identifiers), uploaded as separate file, followed by a Methods and Protocols section in which we encourage the authors to describe their methods using a step-by-step protocol format with bullet points, to facilitate the adoption of the methodologies across labs. More information on how to adhere to this format as well as downloadable templates (.doc or .xls) for the Reagents and Tools Table can be found in our author guidelines (section 'Structured Methods'):

14) Please add up to five keywords to the manuscript and order the manuscript sections like this, using these names:
Title page - Abstract - Keywords - Introduction - Results - Discussion - Methods - Data availability section - Acknowledgements - Disclosure and Competing Interests Statement - References - Figure legends - Expanded View Figure legends

I look forward to seeing a revised version of your manuscript when it is ready. Please let me know if you have questions or comments regarding the revision.

Yours sincerely,

Referee #1:

In the manuscript entitled "RAI14 links mechanical forces to Hippo signaling", Jeong et al. reported retinoic acid-induced protein 14 (RAI14) as a key regulator of the Hippo pathway in mechanotransduction. RAI14 negatively regulated the Hippo pathway by interacting with and translocating NF2 onto F-actin. Reduced mechanical force through actin cytoskeleton (e.g. F-actin depolymerization, soft ECM) promoted the proteasome-dependent degradation of RAI14, which led to the NF2-LATS1 complex formation and Hippo pathway activation. Clinically, RAI14 acted as a potential oncoprotein that was highly expressed in gastric tumors with high mechanical property and required for gastric cancer cell growth. Overall, this study is very interesting, as it provided new mechanistic insights into the Hippo pathway regulation in actin-dependent mechanotransduction. Here, several points were suggested for the authors to further improve their study.

Major points:

1. The role of RAI14 in regulating NF2 and the Hippo pathway needs to be further elucidated. Based on the proposed model, RAI14 bound NF2 to move it onto F-actin, resulting in the reduced the interaction of NF2 with LATS (Fig 8). The authors need to clarify exactly how the NF2-LATS complex is negatively regulated by RAI14. Does RAI14 directly bind NF2 to inhibit its association with LATS? Or does F-actin binding inhibit NF2 to bind LATS?
2. Fig 2D-E showed the very N-terminal 1-16 amino acids were required for NF2 to bind RAI14. This made the NF2 (17-595) mutant a good control for the authors to further confirm their findings. For example, can NF2 (17-595) still bind F-actin? Can NF2 (17-595) have stronger interaction with LATS as compared to wildtype NF2? Can NF2 (17-595) induce higher activity of the Hippo pathway as compared to wildtype NF2?
3. Loss of RAI14 still increased YAP phosphorylation in the NF2 knockdown cells (Fig 3A), which was inconsistent with the authors' working model (Fig 8). Is this due to the remaining NF2 in the knockdown cells? The authors need to use NF2 KO cells or NF2-deficient cell lines (e.g. MDA-MB-231) to address this issue.
4. The authors showed low mechanical force induced the proteasome-dependent degradation of RAI14 via neddylation instead of ubiquitination (Fig 4; EV3D). Although it is unreasonable to ask for the mechanism of RAI14 neddylation (e.g. neddylation site, E3 ligase) at current stage, the authors should at least confirm whether the RAI14 neddylation level is increased in the cells with low mechanical force (e.g. LatB treatment, Soft ECM).
5. Treatment of LATS inhibitor TRULI did not fully rescue the growth of the RAI14 knockdown cells (Fig 7C-D), suggesting the additional Hippo-independent roles of RAI14 in regulating cell proliferation. Therefore, the authors should tune down some of

their related conclusions in the manuscript.

Minor points:

1. There was a slight band shift of RAI14 in the cells with low mechanical force (Fig 4A, B, C, E, F), suggesting potential post-translational modification (PTM) induced for RAI14. What is this PTM? Is this PTM required for the degradation of RAI14 in the cells with low mechanical force?
2. In Fig 6B, RAI14 level was not normalized between the stiff and soft ECM samples, making the data not convincing.
3. In Fig EV1B, is the P value ($P=0.1343$, not significant) in the colon cancer survival data correct?
4. NF2 is known to bind phospholipids on the plasma membrane and some studies reported its nuclear localization. Based on this study, NF2 is also localized on F-actin. The authors are suggested to include more discussions about the spatial localization of NF2 and its related functions.
5. In the end of the section "Destabilization of F-actin induces RAI14 proteasomal degradation" in the manuscript, "Fig. I" should be "Fig. 4I".

Referee #2:

Dr Jho and coworkers identify RAI14 as a novel regulator of YAP/TAZ that by integrating mechanical signals driven by active polymerization and Hippo signaling regulates cell growth.

Evidence suggests that RAI14 is over-expressed and activated in gastric cancer, thereby up-regulating YAP/TAZ activity.

The authors provide sufficient genetic and biochemical evidence to support RAI14 regulation of YAP/TAZ through the modulation of NF2 interaction with F-actin and LATS1.

The pro-oncogenic role of RAI14 in gastric cancer is supported by epidemiological data and in vitro cellular assays.

All the experiments are duly designed and analysed, with the exception of PLA assays for which the authors show some representative pictures but fails to provide a quantitative assessment supported by statistical analysis. Authors need to report n* of PLA foci/cell and perform statistical tests for data displayed in fig 2C and other figures, similarly to what reported for the PLA data shown in fig 5D, 6D,E,F,I.

Minor point:

Amend the following sentence: "we also confirmed that it reduced F-actin formation (Fig. I). "" correct to fig. 4I

Referee #3:

Jeong et al, report that Retinoic acid-induced protein 14 (RAI14), which was identified through a TCGA datasets screening method, is upregulated in gastric cancer (stomach adenocarcinoma) consistent with YAP/TAZ. They find that RAI14 responds to F-actin cues and interacts with the N-terminus of merlin (NF2) in a mechanically enriched environment. Sequestration of NF2 at F-actin regions with increased mechanical tension is suggested to activate YAP/TAZ nuclear and hence transcriptional output. Finally, they also reveal a good clinical association of RAI14 with YAP/TAZ in gastric cancer that is associated with poor prognosis. Overall, this is a potentially interesting and novel work linking RAI14 to NF-2 sequestration with F-actin in conditions corresponding to high rigidity and cytoskeletal tension which is of significance to the YAP/TAZ community. The interaction of RAI14 with NF2, Proximity-ligation assays showing the interaction in stiff matrix and the observation that RAI14 is proteasomal degraded in low mechanical conditions represent the major strengths of this study. However, some concerns related to the mechanobiological impact of Rai14 on YAP attenuate my interests in the findings and therefore, requires additional efforts to make this study more robust for publication.

Major concerns:

1. The effect of YAP nuclear to cytoplasmic expulsion by RAI14 knockdown remains less convincing. The data shown in fig 3B and EV2E clearly suggests that a portion of YAP is increased in the cytosol without much impact on nuclear levels. This is concerning with the subsequent impact on gene expression as nuclear YAP may still continue transcriptional output through TEADs.
2. Fig 1B-C: Despite the observation that Rai14 expression could significantly increase TEAD transcription (luciferase assays), the resulting gene expression changes of Y/T targets remain modest at only 1-1.5 folds increase. This could be due to a basal

high levels of YAP activity and therefore needs to be tested in conditions which have inherently low Y/T or by changing their model systems.

3. Fig 1i- The authors note the destabilization of TAZ upon RAI14 but no effects on YAP stability. Intuitively, if pYAP levels are so high on RAI14 kd, cytoplasmic sequestered YAP is expected to be degraded. However this does not appear to be the case.

4. Does expression of RAI14 (and the ankyrin repeat domain 1-424 as negative control) in low stiffness override the low mechanics to activate nuclear YAP. This is key to support their main observations in a scenario which has low y/t activity as baseline.

Other concerns that need attention:

1. Lack of figure number in the concerned figure panels making it difficult to review

2. Though well written for most parts, the manuscript should be carefully edited (see missing label for 4I on page 8 etc)

3. It is unclear to me why some key experiments on YAP activity and nuclear localization was done in HEK293. Results should be more shown in STAD cells and some of the data shown as EV panels should be in main text particularly if done in STAD cells.

4. Fig 7E-F- A YAP/TAZ target protein should be included in these gastric cancer patient tissues western blots.

5. Fig 7B: the phenotypic relevance of RAI14-YAP on cancer cell survival should be in relevant STAD cell lines and not HEK293A. Also, VP is quite promiscuous as YAP/TEAD inhibitor and hits other TFs. Therefore, these results should be validated with more improved Y/T inhibitors that are recently available.

6. The impact of RAI14 overexpression on STAD cell proliferation in YAP knockdown conditions should be included.

Referee #1:

In the manuscript entitled "RAI14 links mechanical forces to Hippo signaling", Jeong et al. reported retinoic acid-induced protein 14 (RAI14) as a key regulator of the Hippo pathway in mechanotransduction. RAI14 negatively regulated the Hippo pathway by interacting with and translocating NF2 onto F-actin. Reduced mechanical force through actin cytoskeleton (e.g. F-actin depolymerization, soft ECM) promoted the proteasome-dependent degradation of RAI14, which led to the NF2-LATS1 complex formation and Hippo pathway activation. Clinically, RAI14 acted as a potential oncoprotein that was highly expressed in gastric tumors with high mechanical property and required for gastric cancer cell growth. Overall, this study is very interesting, as it provided new mechanistic insights into the Hippo pathway regulation in actin-dependent mechanotransduction. Here, several points were suggested for the authors to further improve their study.

We greatly appreciate the reviewer's thoughtful and positive feedback. We believe that the suggestions are constructive and helpful for enhancing the quality of the manuscript. We have provided a point-by-point response below. In the revised manuscript, the revised content has been marked with a green color.

Major points:

1. The role of RAI14 in regulating NF2 and the Hippo pathway needs to be further elucidated. Based on the proposed model, RAI14 bound NF2 to move it onto F-actin, resulting in the reduced the interaction of NF2 with LATS (Fig 8). The authors need to clarify exactly how the NF2-LATS complex is negatively regulated by RAI14. Does RAI14 directly bind NF2 to inhibit its association with LATS? Or does F-actin binding inhibit NF2 to bind LATS?

Rebuttal Figure 1. Figuring out how NF2-LATS complex is regulated by RAI14

To address the reviewer's comment, we tried to investigate whether the interaction of RAI14 with F-actin is a crucial determinant in the regulation of the NF2-LATS complex or whether RAI14 itself is a critical factor. Since we showed that serum stimulation significantly elevated F-actin polymerization

levels, as previously reported by Yu *et al.* (2012), we proceeded to investigate whether serum stimulation influenced the interaction between RAI14, NF2, and LATS1. Upon serum stimulation, we observed that the interaction between RAI14-NF2 was increased, while the interaction between NF2-LATS1 was decreased (Rebuttal Fig 1A and B). This indicates that F-actin may facilitate the formation of the RAI14-NF2 complex. Next, we asked whether RAI14 alone could regulate the NF2-LATS1 interaction. We found that RAI14 overexpression reduced NF2-LATS1 interaction, suggesting that RAI14 alone could inhibit NF2-LATS1 interaction (Rebuttal Fig 1C). Interestingly, under the condition of F-actin deficiency via treatment of Lat.B, overexpression of RAI14 could still reduce NF2-LATS1 interaction (Rebuttal Fig 1D), suggesting that RAI14 itself can control the interaction between NF2 and LATS1 under the condition of F-actin deficiency. These data are consistent with the data shown in the originally submitted manuscript that a single knockdown of RAI14 increases the NF2-LATS1 complex. Taken together, we would like to propose that NF2-LATS1 complex formation is inhibited by RAI14 binding to NF2 and that this inhibition is promoted by NF2 sequestration via RAI14 to F-actin. We have included the Rebuttal Fig 1A, B, C, and D in the revised manuscript Fig 6I, J, M, and N, respectively. A more comprehensive response to this issue is described in the revised manuscript on pp. 10-11. Thank you for critical suggestions, which allowed us to clarify an important issue.

2. Fig 2D-E showed the very N-terminal 1-16 amino acids were required for NF2 to bind RAI14. This made the NF2 (17-595) mutant a good control for the authors to further confirm their findings. For example, can NF2 (17-595) still bind F-actin? Can NF2 (17-595) have stronger interaction with LATS1 as compared to wildtype NF2? Can NF2 (17-595) induce higher activity of the Hippo pathway as compared to wildtype NF2?

We appreciate the reviewer's suggestion to perform experiments with the NF2 (17-595) mutant form. Since the NF2 (17-595) form has a low ability to interact with RAI14, we wondered if it also has a low

Rebuttal Figure 2. Effect of NF2 (17-595) mutant form in F-actin binding, interaction with LATS1 and activation of Hippo signaling.

ability to bind F-actin, while having a stronger interaction with LATS1 and activation of Hippo signaling.

By performing the F-actin binding assay, we found that the NF2 (17-595) form had significantly reduced binding to F-actin compared to wild-type NF2 (Rebuttal Fig 2A). In addition, the (17-595) form showed increased interaction with LATS1 and phosphorylation of LATS1 compared to wild-type NF2, implying that this mutant form has a greater ability to activate Hippo signaling (Rebuttal Fig 2B). Rebuttal Fig 2A and B have been

added to the revised manuscript as Figures EV4H and I, respectively.

3. Loss of RAI14 still increased YAP phosphorylation in the NF2 knockdown cells (Fig 3A), which was inconsistent with the authors' working model (Fig 8). Is this due to the remaining NF2 in the knockdown cells? The authors need to use NF2 KO cells or NF2-deficient cell lines (e.g. MDA-MB-231) to address this issue.

Rebuttal Figure 3. Effect of RAI14 knockdown in MDA-MB-231 cells.

We agree with the reviewer's comment that loss of RAI14 still induced YAP phosphorylation in the NF2 knockdown cells. We believe that this is because the remaining NF2 is still functioning or that there is an NF2-independent pathway for loss of RAI14 to induce phosphorylation of YAP. To address the reviewer's comment, we used MDA-MB-231 cells, which have been reported to have extremely low expression of NF2 due to a homozygous non-sense mutation of NF2 (Dupont *et al*, 2011). In MDA-MB-231 cells, knockdown of RAI14 alone neither effectively induce YAP phosphorylation (Rebuttal Fig 3A) or downregulate YAP target gene expression (Rebuttal Fig 3B). However, RAI14 knockdown together with NF2 overexpression induced YAP phosphorylation and reduced YAP target gene expression in MDA-MB-231 cells (Rebuttal Fig 3A and B). Although we cannot completely exclude the possibility that RAI14 depletion activates Hippo signaling in an NF2 independent manner, our data suggest that NF2 mainly mediates phosphorylation of YAP and downregulation of YAP target gene expression when the RAI14 levels are reduced. Rebuttal Fig 3A and B have been added to the revised manuscript as Figures 3D and E, respectively.

4. The authors showed low mechanical force induced the proteasome-dependent degradation of RAI14 via neddylation instead of ubiquitination (Fig 4; EV3D). Although it is unreasonable to ask for the

Rebuttal Figure 4. Treatment of Lat.B upregulated neddylation of RAI14.

mechanism of RAI14 neddylation (e.g. neddylation site, E3 ligase) at current stage, the authors should at least confirm whether the RAI14 neddylation level is increased in the cells with low mechanical force (e.g. LatB treatment, Soft ECM).

Based on the reviewer's comment, we expected that RAI14 neddylation would be increased under low mechanical force conditions. To determine the level of RAI14 neddylation, overexpression of HA-NEDD8 and immunoprecipitation of RAI14 were performed. We confirmed that the neddylation level of RAI14 was increased in cells treated with Lat. B (see the ratio of HA(NEDD8) and RAI4 levels. **Rebuttal Fig 4A**). Thanks to the generous consideration of the reviewer, we left the identification of the neddylation site and the E3 ligase of RAI14 as further studies. **Rebuttal Fig 4A** has been added to the revised manuscript **Fig 4H**.

5. Treatment of LATS inhibitor TRULI did not fully rescue the growth of the RAI14 knockdown cells (Fig 7C-D), suggesting the additional Hippo-independent roles of RAI14 in regulating cell proliferation. Therefore, the authors should tune down some of their related conclusions in the manuscript.

We agree with the reviewer's comment. Since TRULI is not sufficient to fully rescue the effect of RAI14 knockdown, our data suggest that there are additional Hippo-independent roles of RAI14 in regulating cell proliferation, although we cannot provide specific mechanisms. At the reviewer's suggestion, we have toned down our conclusions in the manuscript and the edited content is shown below.

Before editing	After editing (highlighted in red)
(Page 10) Treatment with TRULI rescued the shRAI14-mediated reduction in colony-forming ability and cell growth rate (Fig. 7C, D). These results suggest that RAI14 promotes cell proliferation and growth in a Hippo signaling-dependent manner.	(Page 11) Treatment with TRULI significantly, but not completely, rescued the shRAI14-mediated reduction in colony-forming ability and cell growth rate (Fig. 7G, H), suggesting that Hippo signaling is involved in RAI14-mediated cell proliferation and growth regulation, although we cannot

exclude the possibility that there are additional Hippo-independent roles of RAI14 in regulating cell proliferation and growth regulation.

Minor points: 1.

There was a slight band shift of RAI14 in the cells with low mechanical force (Fig 4A, B, C, E, F), suggesting potential post-translational modification (PTM) induced for RAI14. What is this PTM? Is this PTM required for the degradation of RAI14 in the cells with low mechanical force?

Rebuttal Figure 5. Identification of RAI14 PTM.

As the reviewer pointed out, we also observed a band mobility shift of RAI14 at low mechanical force. It is known that protein phosphorylation induces a band mobility shift (Moon et al, 2017). Therefore, we hypothesized that the PTM was phosphorylation of RAI14. Treatment with lambda phosphatase (λ-PPase) reversed the mobility shift of the band (Rebuttal Fig 5A), implying that the mobility shift is due to phosphorylation of RAI14. Analysis of the protein sequence of RAI14 revealed that there is a phosphorylation motif for mitogen-activated protein kinases (MAPKs) and nemo-like kinase (NLK) (Rebuttal Fig 5B), suggesting that MAPKs and NLK may be the kinases for RAI14. We propose that phosphorylation of RAI14 by MAPKs or NLK leads to neddylation of RAI14, resulting in its proteasomal degradation. We leave the verification of the above hypothesis as further studies at this stage, as we believe they are beyond the scope of this manuscript, and ask for the reviewer's generous understanding. Rebuttal Fig 5A has been added to the revised manuscript Fig. EV2E. Rebuttal Fig 5B was shown only in this rebuttal.

2. In Fig 6B, RAI14 level was not normalized between the stiff and soft ECM samples, making the data not convincing.

As suggested by the reviewer, we repeated the experiment in Fig 6B of the previous manuscript and normalized the RAI14 level. In these repeated experiments, we also consistently confirmed that the interaction between RAI14 and NF2 was decreased at low mechanical forces (Rebuttal Fig. 6A). Rebuttal Fig 6A replaces the previous data and has been added to the revised manuscript Fig 6B. Quantification for the ratio of NF2/RAI14 shown in Fig. 6B was presented in Fig EV4A.

Rebuttal Figure 6. Matrix stiffness dependent interaction between RAI14 and NF2.

3. In Fig EV1B, is the P value (P=0.1343, not significant) in the colon cancer survival data correct?

We intended to show that high expression of RAI14 in colorectal cancer patients is associated with poor prognosis. Our mistake in overlooking the P value, which is greater than 0.05, is obvious, and we apologize to the reviewer for our carelessness. As a result, we have decided to exclude the previous Fig. EV1B data from the revised manuscript.

4. NF2 is known to bind phospholipids on the plasma membrane and some studies reported its nuclear localization. Based on this study, NF2 is also localized on F-actin. The authors are suggested to include more discussions about the spatial localization of NF2 and its related functions.

In the Discussion section, we have added the content describing NF2's phospholipid-binding property, nuclear localization and related functions. The added content is shown below.

Before editing
(Page 13) ... and the released NF2 interacts with LATS1, leading to activation of the Hippo signaling pathway.
After editing (highlighted in red)
(Page 15) ... and the released NF2 interacts with LATS1, leading to activation of the Hippo signaling pathway.
The diverse cellular localization of NF2 and its associated function have been investigated. NF2 binds phospholipids, promoting its localization to the plasma membrane and activation of Hippo signaling

(Hong et al, 2020). NF2 is also reported to act in the nucleus. NF2 inhibits the E3 ligase CRL4^{DCAF1}, which ubiquitinates and degrades LATS1/2 in the nucleus. Therefore, NF2 stabilizes LATS1/2 in the nucleus, facilitating the phosphorylation and inhibition of YAP at the nuclear level (Li et al., 2014). Another study showed that the formation of a circumferential actin belt at the apical region of cells promotes the interaction between NF2 and YAP in the nucleus, resulting in export of the complex and suppression of YAP activity (Furukawa et al, 2017). It will be interesting to examine whether changes in RAI14 levels have any effect on the previously known role of NF2 in regulating Hippo signaling.

5. In the end of the section "Destabilization of F-actin induces RAI14 proteasomal degradation" in the manuscript, "Fig. I" should be "Fig. 4I".

We appreciate your review and apologize for the error. The typo has been corrected.

Referee #2:

Dr Jho and coworkers identify RAI14 as a novel regulator of YAP/TAZ that by integrating mechanical signals driven by active polymerization and Hippo signaling regulates cell growth.

Evidence suggests that RAI14 is over-expressed and activated in gastric cancer, thereby up-regulating YAP/TAZ activity.

The authors provide sufficient genetic and biochemical evidence to support RAI14 regulation of YAP/TAZ through the modulation of NF2 interaction with F-actin and LATS1.

The pro-oncogenic role of RAI14 in gastric cancer is supported by epidemiological data and in vitro cellular assays.

All the experiments are duly designed and analysed, with the exception of PLA assays for which the authors show some representative pictures but fails to provide a quantitative assessment supported by statistical analysis. Authors need to report n* of PLA foci/cell and perform statistical tests for data displayed in fig 2C and other figures, similarly to what reported for the PLA data shown in fig 5D, 6D,E,F,I.

We sincerely appreciate the reviewer's positive assessments and comments. As suggested by the reviewer, we have performed quantitative assessments in Fig 2C, EV4D, E, F, and G. The number of PLA foci per cell and statistical tests have been included in the figure itself and in the figure legends. Rebuttal Fig 1A, B, C, D, and E correspond to previous Fig 2C, EV4D, E, F, and G, respectively. In the revised manuscript, we have added new PLA data using AGS and SNU638 gastric cancer cells. Quantification and statistical tests were also performed on the new data (Rebuttal Fig 1F and G). Rebuttal Fig 1A, B, C, D, and E replace the previous data and have been added to the revised manuscript as Fig 2C, EV3D, E, F, and G, respectively. Rebuttal Fig 1F and G have been added to the revised manuscript Fig 2D and E.

Rebuttal Figure 1. Quantification and statistical analysis of PLA data

Minor point:

Amend the following sentence: "we also confirmed that it reduced F-actin formation (Fig. I). "" correct to fig. 4I

We appreciate your review and apologize for the error. The typo has been corrected.

Referee #3:

Jeong et al, report that Retinoic acid-induced protein 14 (RAI14), which was identified through a TCGA datasets screening method, is upregulated in gastric cancer (stomach adenocarcinoma) consistent with YAP/TAZ. They find that RAI14 responds to F-actin cues and interacts with the N-terminus of merlin (NF2) in a mechanically enriched environment. Sequestration of NF2 at F-actin regions with increased mechanical tension is suggested to activate YAP/TAZ nuclear and hence transcriptional output. Finally, they also reveal a good clinical association of RAI14 with YAP/TAZ in gastric cancer that is associated with poor prognosis. Overall, this is a potentially interesting and novel work linking RAI14 to NF-2 sequestration with F-actin in conditions corresponding to high rigidity and cytoskeletal tension which is of significance to the YAP/TAZ community. The interaction of RAI14 with NF2, Proximity-ligation assays showing the interaction in stiff matrix and the observation that RAI14 is proteasomal degraded in low mechanical conditions represent the major strengths of this study. However, some concerns related to the mechanobiological impact of Rai14 on YAP attenuate my interests in the findings and therefore, requires additional efforts to make this study more robust for publication.

We greatly appreciate the thoughtful and positive responses of the reviewer. By reflecting on the careful comments, we have tried to solidify and improve our study. Please see our response attached below.

Major concerns:

1. The effect of YAP nuclear to cytoplasmic expulsion by RAI14 knockdown remains less convincing. The data shown in fig 3B and EV2E clearly suggests that a portion of YAP is increased in the cytosol without much impact on nuclear levels. This is concerning with the subsequent impact on gene expression as nuclear YAP may still continue transcriptional output through TEADs.

In accordance with the reviewer's comments, we conducted a meticulous immunofluorescence assay, including a quantification of the nuclear/cytoplasmic YAP ratio and the normalization of nuclear YAP levels to the nuclear DAPI signal. The results demonstrated that RAI14 knockdown led to a statistically significant reduction in both the nuclear/cytoplasmic YAP ratio and the nuclear YAP level (Rebuttal Fig 1A and B). Nevertheless, despite the knockdown of RAI14, some cells still retain nuclear YAP. Despite our best efforts, it proved challenging to establish conditions that would fully exclude nuclear YAP by RAI14 knockdown. This may be due to the involvement of multiple factors in the regulation of YAP localization, and a single knockdown of RAI14 may not be sufficient to completely exclude nuclear YAP. The cell lines or tissues utilized, in conjunction with the experimental setting, also influence the localization of YAP. Therefore, we believe that the partial effect on YAP localization upon RAI14 knockdown is somewhat reasonable. Nevertheless, our quantification data, shown in Rebuttal Fig 1A and B, strongly support the idea that RAI14 knockdown affects the export of nuclear YAP to the cytoplasm. Rebuttal Fig 1A and B replace the previous data and have been added to the revised

manuscript as Fig 1L and 3B, respectively.

Rebuttal Figure 1. Knockdown of RAI14 reduced nuclear/cytoplasmic YAP ratio and nuclear YAP level itself.

2. Fig 1B-C: Despite the observation that Rai14 expression could significantly increase TEAD transcription (luciferase assays), the resulting gene expression changes of Y/T targets remain modest at only 1-1.5 folds increase. This could be due to a basal high levels of YAP activity and therefore needs to be tested in conditions which have inherently low Y/T or by changing their model systems.

Rebuttal Figure 2. Overexpression of RAI14 upregulates expression of YAP target genes.

We concur with the reviewer's assessment that the observed changes in gene expression remained relatively modest in comparison to those observed in reporter assays. In order to maintain a low basal level of YAP activity, we maintained a high level of cell confluency. Under these conditions, we observed that overexpression of RAI14 resulted in the upregulation of YAP target genes by approximately twofold (Rebuttal Fig 2A). We hope that the degree of upregulation meets the reviewer's expectations.

The previously submitted data have been replaced with Rebuttal Fig 2A, which has been added into the revised manuscript as Fig 1C.

3. Fig 1i- The authors note the destabilization of TAZ upon RAI14 but no effects on YAP stability. Intuitively, if pYAP levels are so high on RAI14 kd, cytoplasmic sequestered YAP is expected to be degraded. However this does not appear to be the case.

It is well known that activation of Hippo signaling leads to phosphorylation of YAP and TAZ, resulting in cytoplasmic retention and proteasomal degradation. Therefore, phosphorylation of YAP and TAZ should lead to their destabilization. However, this depends mainly on the context. For example, HEK293A cells,

which are widely used for Hippo signaling experiments, typically showed the destabilization of TAZ but not YAP under the condition of activation of Hippo signaling and induction of their phosphorylation (top left figure (Meng et al., 2015)). We believe this is the reason why phosphorylation of YAP did not show its destabilization in our data.

To further substantiate our findings, we employed another cell line, SNU638, to confirm that knockdown of RAI14 induced not only phosphorylation of YAP but also its destabilization (**Rebuttal Fig 3A**). We hope that our explanation and data will satisfy the reviewer. Rebuttal Fig 3A has been added as Fig 1K in the revised manuscript.

Figures from previous paper (Fig 1b and d) (Meng Z et al, Nat comm, 2015)

Rebuttal Figure 3. Knockdown of RAI14 in SNU638 cells downregulated YAP protein level.

4. Does expression of RAI14 (and the ankyrin repeat domain 1-424 as negative control) in low stiffness override the low mechanics to activate nuclear YAP. This is key to support their main observations in a scenario which has low y/t activity as baseline.

To identify the cellular localization of YAP, we performed an immunofluorescence assay and examined whether overexpression of RAI14 promoted nuclear localization of YAP in both stiff and soft matrix conditions. In both stiff and soft matrix conditions, the cells overexpressing wild-type RAI14 (marked with arrow) increased the nuclear localization of YAP, whereas the cells overexpressing the RAI14 (1-424) form (marked with arrow) did not promote the nuclear localization of YAP (**Rebuttal Fig 4A**). This suggests that overexpression of RAI14 overrides the low mechanics to promote nuclear localization of YAP, and its ankyrin repeat domain, which does not interact with NF2, does not override. To further substantiate our claim, we examined whether wild-type RAI14 or RAI14 (1-424) could rescue the reduced transcriptional activity of the YAP reporter in the low stiffness condition induced by Lat.B treatment. We found that overexpression of RAI14 partially rescued reporter activity, but not in the form of RAI14 (1-424) (**Rebuttal Fig 4B**). Since incubation of cells on low stiffness induces the suppression of YAP transcriptional activity through various pathways in a Hippo-dependent and -independent manner, we believe that this is the reason why partial, but not complete, rescue was observed by RAI14 overexpression. Taken together, our data support the idea that overexpression of RAI14 under low

stiffness conditions leads to nuclear localization of YAP and increased expression of YAP target genes. Rebuttal Fig 4A and B have been added to the revised manuscript as Fig EV4E and F, respectively.

Rebuttal Figure 4. Expression of RAI14 in low stiffness condition partially rescue YAP activity.

Other concerns that need attention:

1. Lack of figure number in the concerned figure panels making it difficult to review

We apologize for this inconvenience. We have added the figure numbers in the revised manuscript.

2. Though well written for most parts, the manuscript should be carefully edited (see missing label for 4I on page 8 etc)

Thank you for your careful review of our manuscript. We have corrected the typos and carefully proofread the whole manuscript.

3. It is unclear to me why some key experiments on YAP activity and nuclear localization was done in HEK293. Results should be more shown in STAD cells and some of the data shown as EV panels should be in main text particularly if done in STAD cells.

We agree with the reviewer's comment that key experiments should have been performed in STAD-related cell lines. We have added a RAI14 knockdown experiment in SNU638 cells (Rebuttal Fig 3A, Fig 1K in the revised manuscript) and RAI14 overexpression with reporter assay in SNU638 cells (Rebuttal Fig 5A). To confirm the endogenous interaction between RAI14 and NF2 in AGS and SNU638 cells, a PLA assay was performed and RAI14-NF2 PLA signals were detected in AGS and SNU638 cells (Rebuttal Fig 5B and C). In the initially submitted, we also performed some key experiments using STAD-related cell lines. Fig EV2A and D in the previous manuscript have been moved to Fig 1B and I in the revised manuscript. Rebuttal Fig 5A, B and C have been added to the revised manuscript Fig 1B, 2D and 2E, respectively.

4. Fig 7E-F- A YAP/TAZ target protein should be included in these gastric cancer patient tissues western blots.

In accordance with the reviewer's comment, we sought to examine the levels of a well-known YAP target protein, CTGF, in patients with diffuse and intestinal-type gastric cancer. Our findings revealed that CTGF levels were upregulated in both types of gastric cancer patients (Rebuttal Fig 6A, left panel). As with the expression pattern of RAI14, YAP, TAZ, and F-actin, quantification of western bands reveals that the upregulation pattern of CTGF was more significant in diffuse-type patients (Rebuttal Fig 6A, right panel). This figure has been added to the revised manuscript as Fig EV5F.

Rebuttal Figure 6. Detection of CTGF western bands in diffuse and intestinal-type gastric cancer patients

5. Fig 7B: the phenotypic relevance of RAI14-YAP on cancer cell survival should be in relevant STAD cell lines and not HEK293A. Also, VP is quite promiscuous as YAP/TEAD inhibitor and hits other TFs. Therefore, these results should be validated with more improved Y/T inhibitors that are recently available.

We agree with the reviewer's comment that VP has side effects besides TEAD inhibition. Therefore, an alternative TEAD inhibitor, MGH-CP1, which inhibits TEAD palmitoylation (Li et al., 2020), was employed. Furthermore, a colony-forming assay and cell growth assay were conducted using AGS and SNU638 cell lines, as well as HEK293A cells. The results demonstrated that overexpression of RAI14 upregulated colony-forming capacity in AGS, SNU638, and HEK293A cells. This effect was blocked by treatment with MGH-CP1 (Rebuttal Fig 7A-C). As with the colony-forming assay, the same patterns were observed in the cell growth assay (Rebuttal Fig 7D-F). These data provide further support for our conclusion that RAI14 promotes cell proliferation and growth by regulating Hippo signaling. Rebuttal Fig 7A and D have been added to the revised manuscript Fig 7D and C, respectively. Rebuttal Fig 7B and E have been added to the revised manuscript Fig 7F and E, respectively. Rebuttal Fig 7C and F have been included in this rebuttal only.

Rebuttal Figure 7. Effect of RAI14 overexpression and treatment of TEAD inhibitor MGH-CP1 in cell proliferation and growth.

6. The impact of RAI14 overexpression on STAD cell proliferation in YAP knockdown conditions should be included.

In accordance with the reviewer's comment, we performed RAI14 overexpression and YAP/TAZ knockdown in AGS, SNU638, and HEK293A cells and followed up with colony forming assay and cell growth assay. Similar to the effects of MGH-CP1, the effects of RAI14 overexpression were blocked by YAP/TAZ knockdown in AGS, SNU638 and HEK293A cells (Rebuttal Fig 8). These data strengthen our conclusions that RAI14 promotes cell proliferation and growth by regulating YAP/TAZ activity. Rebuttal Fig 8 has been added to the revised manuscript Appendix Fig 3.

Rebuttal Figure 8. Effect of RAI14 overexpression and YAP/TAZ knockdown in cell proliferation and growth

References

- Dupont S, Morsut L, Aragona M, Enzo E, Giulitti S, Cordenonsi M, Zanconato F, Le Digabel J, Forcato M, Bicciato S *et al* (2011) Role of YAP/TAZ in mechanotransduction. *Nature* 474: 179-183
- Furukawa KT, Yamashita K, Sakurai N, Ohno S (2017) The Epithelial Circumferential Actin Belt Regulates YAP/TAZ through Nucleocytoplasmic Shuttling of Merlin. *Cell reports* 20: 1435-1447
- Hong AW, Meng Z, Plouffe SW, Lin Z, Zhang M, Guan KL (2020) Critical roles of phosphoinositides and NF2 in Hippo pathway regulation. *Genes & development* 34: 511-525
- Li Q, Sun Y, Jarugumilli GK, Liu S, Dang K, Cotton JL, Xiol J, Chan PY, DeRan M, Ma L *et al* (2020) Lats1/2 Sustain Intestinal Stem Cells and Wnt Activation through TEAD-Dependent and Independent Transcription. *Cell Stem Cell* 26: 675-692 e678
- Li W, Cooper J, Zhou L, Yang C, Erdjument-Bromage H, Zagzag D, Snuderl M, Ladanyi M, Hanemann CO, Zhou P *et al* (2014) Merlin/NF2 loss-driven tumorigenesis linked to CRL4(DCAF1)-mediated inhibition of the hippo pathway kinases Lats1 and 2 in the nucleus. *Cancer cell* 26: 48-60
- Meng Z, Moroishi T, Mottier-Pavie V, Plouffe SW, Hansen CG, Hong AW, Park HW, Mo JS, Lu W, Lu S *et al* (2015) MAP4K family kinases act in parallel to MST1/2 to activate LATS1/2 in the Hippo pathway. *Nature communications* 6: 8357
- Moon S, Kim W, Kim S, Kim Y, Song Y, Bilousov O, Kim J, Lee T, Cha B, Kim M *et al* (2017) Phosphorylation by NLK inhibits YAP-14-3-3-interactions and induces its nuclear localization. *EMBO Rep* 18: 61-71
- Yin F, Yu J, Zheng Y, Chen Q, Zhang N, Pan D (2013) Spatial organization of Hippo signaling at the plasma membrane mediated by the tumor suppressor Merlin/NF2. *Cell* 154: 1342-1355
- Yu FX, Zhao B, Panupinthu N, Jewell JL, Lian I, Wang LH, Zhao J, Yuan H, Tumaneng K, Li H *et al* (2012) Regulation of the Hippo-YAP pathway by G-protein-coupled receptor signaling. *Cell* 150: 780-791

Dear Prof. Jho,

Thank you for the submission of your revised manuscript to our editorial offices. I have now received the reports from the two referees that I asked to re-evaluate the study, you will find below. As you will see, the referees now fully support the publication of the study in EMBO reports.

Before we can proceed with formal acceptance, I have these editorial requests I ask you to address in a final revised manuscript:

- I would suggest this more comprehensive title:

Retinoic acid-induced protein RAI14 links mechanical forces to Hippo signaling

- Please provide (also) an institutional e-mail address for co-corresponding author In-Seob Lee on the title page of the manuscript text file.

- Please remove the section numbering from the Methods part.

- Please provide the Appendix file as .pdf with the final updated title on the title page.

- Please carefully check the final figures again that the number "n" for how many independent experiments were performed, their nature (biological versus technical replicates), the bars and error bars (e.g. SEM, SD) and the test used to calculate p-values is indicated in the respective figure legends. Please also check that all the p-values are explained in the legend, and that these fit to those shown in the figure. Please provide statistical testing where applicable. Please avoid the phrase 'independent experiment', but clearly state if these were biological or technical replicates. Please also indicate (e.g. with n.s.) if testing was performed, but the differences are not significant. In case n=2, please show the data as separate datapoints without error bars and statistics. See also:

<http://www.embopress.org/page/journal/14693178/authorguide#statisticalanalysis>

If n<5, please show single datapoints for diagrams. Moreover:

- Please note that the exact p values are not provided in the legend of figure EV 5e.

- Please indicate the statistical test used for data analysis in the legends of figures 1b-h, j, l-n; 2c-e, i-j; 3a-c, e; 4b-c, e-i; 5e; 6d-e, g; 7a-h, k; EV 1a-c; EV 2b, d; EV 3d-g; EV 4a-d, f-g; EV 5a-d, f, h-k.

- Please note that the white arrowheads are not defined in the legend of figure 5a. This needs to be rectified.

- Please note that the white arrows are not defined in the legend of figure EV 4e. This needs to be rectified.

- Please add to each legend (main, EV and Appendix figures, where applicable) a 'Data Information' section explaining the statistics used or providing information regarding replicates and scales. See:

Such a section is already present in some legends. Just put 'Data Information:' in front in these cases.

- Please use the correct nomenclature for the figures in the Expanded View Figure Legend. The names should be "Figure EVx" instead of "Expanded View Figure x".

- The originally provided Figure EV1 couldn't be converted to PDF by the system. Please resupply this figure as a properly formatted .tif file.

- Please add scale bars of similar style and thickness to all microscopic images (also those in the Appendix), using clearly visible black or white bars (depending on the background). Please place these in the lower right corner of the images themselves. Please do not write on or near the bars in the image but define the size in the respective figure legend. Presently, several scale bars are too thin. Please check.

- Please make sure that all the funding information is also entered into the online submission system and that it is complete and similar to the one in the acknowledgement section of the manuscript text file. Presently the Brain Korea 21 program, sponsored research funds provided by Mr. Hyeonho Kim (perhaps also Asan Bio-Resource Center, Korea Biobank Network [2023-16(267)]) are missing in the submission system. Please check.

- Please remove the reagents tables (S1-S4) from the Appendix but provide this information in a 'Reagents and Tools' table. I have attached a template for that in word format. Please upload the filled in table to the manuscript tracking system as 'Reagent Table' file. Please also adjust all callouts for the previous Appendix tables to this table. The example linked below shows how the table will display in the published article and includes examples of the type of information that should be provided for the different categories of reagents and tools. Please list your reagents/tools using the categories provided in the template and do

not add additional subheadings to the table. Reagents/tools that do not fit in any of the specific categories can be listed under "Other":

https://www.embopress.org/pb%2Dassets/embo-site/msb_177951_sample_FINAL.pdf

- Thank you for providing the requested source data. Please upload this as one folder per figure (with all files for one figure in one folder and ZIPed).

In addition, I would need from you:

Best,

Referee #1:

The authors have fully addressed my questions raised before. The revised manuscript has been significantly improved.

Referee #3:

Happy with the revisions.

The authors addressed the remaining editorial issues.

Prof. Eek-hoon Jho
University of Seoul
Department of Life Science
163 Seoulsiripdaero, Dongdaemun-gu
Seoul, Seoul 02504
Korea, Republic of

Dear Prof. Jho,

I am very pleased to accept your manuscript for publication in the next available issue of EMBO reports. Thank you for your contribution to our journal.

Yours sincerely,
